



# Sensitivity of inverse glacial isostatic adjustment estimates over Antarctica

Matthias O. Willen[1], Martin Horwath[1], Ludwig Schröder[1,3], Andreas Groh[1], Stefan R. M. Ligtenberg[2], Peter Kuipers Munneke[2], and Michiel R. van den Broeke[2]

[1]Technische Universität Dresden, Institut für Planetare Geodäsie, Dresden, Germany
[2]Utrecht University, Institute for Marine and Atmospheric research Utrecht (IMAU), Utrecht, The Netherlands
[3]now at Alfred Wegener Institute, Helmholtz Centre for Polar and Marine Research, Bremerhaven, Germany

**Correspondence:** Matthias Oskar Willen (matthias.willen@tu-dresden.de)

**Abstract.** Glacial isostatic adjustment (GIA) is a major source of uncertainty in estimated ice and ocean mass balance that are based on satellite gravimetry. In particular over Antarctica the gravimetric effect of cryospheric mass change and GIA are of the same order of magnitude. Inverse estimates from geodetic observations are promising for separating the two superimposed mass signals. Here, we investigate the combination of satellite gravimetry and altimetry and how the choice of input data
sets and processing details affect the inverse GIA estimates. This includes the combination for almost full GRACE lifespan (2002-04/2016-08). Further we show results from combining data sets on time-series level. Specifically on trend level, we assess the spread of GIA solutions that arises from (1) the choice of different degree-1 and $C_{20}$ products, (2) different surface elevation change products derived from different altimetry missions and associated to different time intervals, and (3) the uncertainty of firn-process models. The decomposition of the total-mass signal into the ice-mass signal and the apparent GIA-
mass signal depends strongly on correcting for apparent biases in initial solutions by forcing the mean GIA and GRACE trend over the low precipitation zone of East Antarctica to be zero. Prior to bias correction, the overall spread of total-mass change and apparent GIA-mass change using differing degree-1 and $C_{20}$ products is 68 and 72 Gt a$^{-1}$, respectively, for the same time period (2003-03/2009-10). The bias correction suppresses this spread to 6 and 5 Gt a$^{-1}$, respectively. We characterise the firn-process model uncertainty empirically by analysing differences between two alternative surface-mass-balance products. The
differences propagate to a 21 Gt a$^{-1}$ spread in apparent GIA-mass-change estimates. The choice of the altimetry product poses the largest uncertainty on debiased mass-change estimates. The overall spread of debiased GIA-mass change amounts to 18 and 49 Gt a$^{-1}$ for a fixed time period (2003-03/2009-10) and various time periods, respectively. Our findings point out limitations associated with data processing, correction for apparent biases, and time dependency.

## 1 Introduction

The quantification of recent and current sea-level changes plays a crucial role for local, regional, and global projections. Mass changes of the Greenland and Antarctic ice sheets are responsible for approximately 20 % of the global mean sea-level rise between 1991–2010 (Church et al., 2013).




The mass balance of an ice sheet is the difference of surface mass balance (SMB) and ice discharge. It can be determined with several methods (Shepherd et al., 2012, 2018). In one such method, space gravimetry observes temporal gravity changes which result from mass redistribution on Earth. Ice-mass-trend estimation is done with the time-variable gravity fields from the Gravity Recovery And Climate Experiment (GRACE) mission (e.g., Groh et al., 2014; Forsberg et al., 2017) and will be continued by its follow-on mission GRACE-FO.

However, large uncertainty in the ice-mass-change estimates derived from space gravimetry is related to viscoelastic deformation of the solid Earth by glacial isostatic adjustment (GIA). This is the deformation of the solid Earth due to loading variations through glaciation and deglaciation for the last hundreds to thousands of years. Ice-sheet and GIA-mass change signals are superimposed and are of the same order of magnitude over Antarctica (Sasgen et al., 2017). This makes it unavoidable to consider GIA carefully when determining ice-mass change. Moreover, quantified GIA provides insights into the glacial

history of ice sheets or changing tectonic stress (Johnston et al., 1998).

One approach to determine the GIA signal is forward-modelling (e.g. Ivins and James, 2005). GIA forward models are obtained using assumptions about the ice-load history and the solid-Earth rheology, which are both subject to large uncertainties (Whitehouse, 2018; Whitehouse et al., 2019). GIA-induced vertical bedrock elevation change (BEC) derived from Global Navigation Satellite System (GNSS) observations have been used to constrain forward models (e.g., King et al., 2010; Ivins

et al., 2013; Whitehouse et al., 2012) or, more recently, to test probabilistic information of a suite of forward models (Caron et al., 2018).

In an alternative approach, satellite gravimetry and altimetry are combined to separate the GIA and ice-related mass signals (Wahr et al., 2000). Both spaceborne techniques observe a superposition of GIA and ice-sheet-change signals. The combination requires assumptions about the relation between surface-geometry changes and gravity-field changes induced by GIA, and

likewise, between the respective changes induced by ice-sheet processes. These relations may be expressed in terms of effective densities. This combination approach was first implemented by Riva et al. (2009) and later refined by Groh et al. (2012) and Gunter et al. (2014). Hereinafter they are called *inverse* (Whitehouse, 2018) because they use present-day observations to determine the GIA signal (in contrast to forward models). Results from Riva et al. (2009) fit better with GNSS-derived GIA rates than forward models (Thomas et al., 2011).

Recent studies separate the individual processes of the ice sheet and the underlying bedrock with statistical modelling (Zammit-Mangion et al., 2015; Martín-Español et al., 2016a). They use spatial and temporal *a priori* information (from numerical simulations), additional GNSS observations, and altimetry data of several satellite missions. Furthermore, a joint inversion has been presented that takes into account the rheological parameters of the solid Earth (Sasgen et al., 2017). Engels et al. (2018) use a regularised parameter estimation approach (dynamic patch approach) to resolve the superimposed mass trends in

Antarctica. Martín-Español et al. (2016b) compared available GIA solutions from forward modelling and inverse estimation and have shown that differences are beyond indicated uncertainties.

We analyse the sensitivity of inverse GIA estimation towards data input and methodological choices and thereby identify possible causes of discrepancies and attribute the uncertainties. Our inverse GIA estimation is based on the approach of Gunter et al. (2014), but using different and updated data sets. Special attention is paid to firn processes, namely SMB and the volume



change of the firn layer. In inverse GIA estimation, changes in the firn layer overlaying the ice sheet need to be separated from those in the ice layer below. For that purpose, SMB as well as volume change from the firn layer are needed. These are usually provided by regional climate models like RACMO2 (van Wessem et al., 2018), and firn densification models (FDM) forced with these climate models, like IMAU-FDM (Ligtenberg et al., 2011). Uncertainties of model products are poorly known. Here, we characterise the uncertainty by comparing the RACMO2.3p2 SMB product and the SMB from the MAR model result

(Agosta et al., 2019).

Another focus is on the use of ice altimetry data. Different altimeter missions as Envisat, ICESat or CryoSat-2 use different observation techniques and differ in their spatial and temporal coverage. The Multi-Mission (MM) altimetry data set by Schröder et al. (2019) is well suited for a GIA inversion over almost the full GRACE observation period (2002-04/2016-08). The effect of using different gravity-field solutions from the GRACE processing centres and different filtering options is shown

by Gunter et al. (2014). We use different degree-1 and $C_{20}$ products to quantify their effect on inverse GIA estimation. In addition, we demonstrate the combination on time-series level as a generalisation of the combination of linear trends.

Section 2 derives and describes in detail the combination approach, bias correction, and filtering. Afterwards, it is explained how the errors for the firn-process models are characterised and how the sensitivity analysis is performed. Furthermore, the approach is adapted to enable the combination on time-series level. Section 3 describes the used products, processing steps, and

additional assumptions. Section 4 presents results of derived uncertainties of the firn-process models, the sensitivity analysis, and the combination on time-series level. Finally, the results are discussed and the most important findings are summarised in the conclusions.

## 2   Methods

### 2.1   Combination approach

Wahr et al. (2000) were the first to suggest the combination of satellite geodetic methods – gravimetry and altimetry – to estimate GIA. We use the analytical approach from Wahr et al. (1998) to explain gravity changes by mass changes projected into a spherical layer (with radius $a$) – termed area-density changes (ADC) or surface-density changes. Note that a change of mass is with respect to a reference mass distribution. Based on GRACE solutions given in the spherical-harmonic domain, the conversion of changes in Stokes coefficients with degree $n$ and order $m$ ($\Delta c_{nm}$) into spherical harmonic coefficients of ADC

($\Delta \kappa_{nm}$) is

$$\Delta \kappa_{nm} = \frac{2n+1}{1+k'_n} \frac{M_E}{4\pi a^2} \Delta c_{nm}, \tag{1}$$

where $M_E$ is the total mass of the Earth, $a$ the equatorial radius of the reference ellipsoid, and $k'_n$ the second load Love number to account for the deformation potential of the solid Earth induced by the mass redistribution. The linear ADC $\dot{\kappa}_{nm}$ is synthesised into spatial domain $\dot{m}_{\mathrm{grav}}$, which is the superposition of the ADC through GIA, and processes in the ice (ID) and

firn layer

$$\dot{m}_{\mathrm{grav}} = \dot{m}_{\mathrm{GIA}} + \dot{m}_{\mathrm{ID}} + \dot{m}_{\mathrm{firn}}. \tag{2}$$





Note that $\dot{m}_{\mathrm{GIA}}$ is not the GIA-induced mass trend: it is the apparent ADC because of the GIA-induced gravity-field changes. With ID all processes are summarised which are weighted with ice density, e.g. ice-dynamic flow or basal melt. We summarise the ice-induced, or cryospheric, area-density trend as $\dot{m}_{\mathrm{ice}} = \dot{m}_{\mathrm{ID}} + \dot{m}_{\mathrm{firn}}$.

Analogously, the linear surface elevation change (SEC) derived from altimetry $\dot{\tilde{h}}_{\mathrm{alt}}$ is the sum of the linear SEC through ID, firn, GIA, and elastic BEC

$$\dot{\tilde{h}}_{\mathrm{alt}} = \dot{h}_{\mathrm{GIA}} + \dot{h}_{\mathrm{elastic}} + \dot{h}_{\mathrm{ID}} + \dot{h}_{\mathrm{firn}}. \tag{3}$$

Note that GIA refers to the viscoelastic deformation of the solid Earth. The elastic BEC ($\dot{h}_{\mathrm{elastic}}$) through present-day ice-mass changes is reduced prior to the combination by defining $\dot{h}_{\mathrm{alt}} = \dot{\tilde{h}}_{\mathrm{alt}} - \dot{h}_{\mathrm{elastic}}$.

The process-related elevation and area-density changes are linked with effective density assumptions ($\rho_{\mathrm{GIA}}, \rho_{\mathrm{ID}}$)

$$\dot{m}_{\mathrm{GIA}} = \rho_{\mathrm{GIA}} \cdot \dot{h}_{\mathrm{GIA}} \tag{4}$$

$$\dot{m}_{\mathrm{ID}} = \rho_{\mathrm{ID}} \cdot \dot{h}_{\mathrm{ID}}. \tag{5}$$

Rearranging Eq. (3)

$$\dot{h}_{\mathrm{ID}} = \dot{h}_{\mathrm{alt}} - \dot{h}_{\mathrm{firn}} - \dot{h}_{\mathrm{GIA}} \tag{6}$$

and substituting it together with Eq. (4) and (5) into Eq. (2) leads to

$$\dot{m}_{\mathrm{grav}} = \rho_{\mathrm{GIA}} \dot{h}_{\mathrm{GIA}} + \rho_{\mathrm{ID}}(\dot{h}_{\mathrm{alt}} - \dot{h}_{\mathrm{firn}} - \dot{h}_{\mathrm{GIA}}) + \dot{m}_{\mathrm{firn}}, \tag{7}$$

which can be solved for

$$\dot{h}_{\mathrm{GIA}} = \frac{\dot{m}_{\mathrm{grav}} - \rho_{\mathrm{ID}}(\dot{h}_{\mathrm{alt}} - \dot{h}_{\mathrm{firn}}) - \dot{m}_{\mathrm{firn}}}{\rho_{\mathrm{GIA}} - \rho_{\mathrm{ID}}}. \tag{8}$$

In Gunter et al. (2014), Eq. (8) is modified with a criterion to include assumptions about the difference $\dot{h}_{\mathrm{alt}} - \dot{h}_{\mathrm{firn}}$ by *a priori* uncertainties. $\rho_{\mathrm{ID}}$ is replaced by $\rho_\alpha$ to permit the following case distinction:

$$\dot{h}_{\mathrm{GIA}} = \frac{\dot{m}_{\mathrm{grav}} - \rho_\alpha(\dot{h}_{\mathrm{alt}} - \dot{h}_{\mathrm{firn}}) - \dot{m}_{\mathrm{firn}}}{\rho_{\mathrm{GIA}} - \rho_\alpha} \tag{9}$$

where

$$\rho_\alpha = \begin{cases} \rho_{\mathrm{ID}}, & \text{if } \dot{h}_{\mathrm{alt}} - \dot{h}_{\mathrm{firn}} < 0 \\ & \text{and } |\dot{h}_{\mathrm{alt}} - \dot{h}_{\mathrm{firn}}| > 2\sigma_h \\ \rho_{\mathrm{firn}}, & \text{if } \dot{h}_{\mathrm{alt}} - \dot{h}_{\mathrm{firn}} > 0 \\ & \text{and } |\dot{h}_{\mathrm{alt}} - \dot{h}_{\mathrm{firn}}| > 2\sigma_h \\ 0, & \text{otherwise} \end{cases} \tag{10}$$



with

$$\sigma_h = \sqrt{\sigma_{\dot{h}_{\mathrm{alt}}}^2 + \sigma_{\dot{h}_{\mathrm{firn}}}^2} \tag{11}$$

The case distinction is made to account for uncertainties in altimetry and the firn densification model (FDM) by using *a priori* knowledge on ice-sheet processes. The GIA-induced BEC is in the millimetre per year range, whereas $\dot{h}_{\mathrm{firn}}$ and $\dot{h}_{\mathrm{ID}}$ can be in the centimetre to meter per year range. If altimetry and FDM are perfect, $\dot{h}_{\mathrm{alt}} - \dot{h}_{\mathrm{firn}}$ would leave essentially $\dot{h}_{\mathrm{ID}}$ (apart from a

very small $\dot{h}_{\mathrm{GIA}}$). The following case distinction is made: If the altimetry-derived SEC is significantly more negative than SEC from the FDM, an ice-dynamic-induced SEC is assumed (glacial thinning). Gunter et al. (2014) argue that only one region in Antarctica is known to show glacial thickening. The area of the Kamb Ice Stream is therefore treated separately. For case II in Eq. (10) it is assumed that the FDM underestimates SEC due to firn processes and the remaining part therefore must not be weighted with ice density but with firn density. If the difference is not significant (smaller than $2\sigma_h$), it is not considered (case III

in Eq. 10). This approach has the advantage to solve for GIA without a predefined spatial mask to distinguish between firn and ice processes (e.g. density mask in Riva et al. (2009)) except for regions with ice-dynamic thickening. An underestimated $\sigma_h$ leads to differences between $\dot{h}_{\mathrm{alt}}$ and $\dot{h}_{\mathrm{firn}}$ being included in the mass balance, although they may not be significant. An overestimated $\sigma_h$ will likely lead to case III in Eq. (10), also for significant signals. In this case, data of altimetry and the model information of the FDM are not taken into account – but $\dot{m}_{\mathrm{firn}}$ and $\dot{m}_{\mathrm{grav}}$ will be still fully used.

## 2.2   Bias correction

To investigate the combination methodology, one of our aims is to exactly reproduce the method of Gunter et al. (2014). The estimation of (1) the GIA-induced BEC and (2) the mass balance is performed in a sequence. Gunter et al. (2014) crucially introduce two bias corrections to consider offsets introduced e.g. by systematic errors in degree-1 and $C_{20}$. They argue that the effect of such offsets are significantly larger than potential mass signals in a low precipitation zone (LPZ) of the East Antarctic

Ice Sheet.

First, the *LPZ-based GIA bias correction* $\bar{\dot{h}}_{\mathrm{GIA,LPZ}}$ is applied. It is assumed that the GIA-induced BEC should be negligibly small in this area. A remaining signal in the GIA estimate is interpreted as a bias. Therefore the mean GIA-induced BEC within the LPZ $\bar{\dot{h}}_{\mathrm{GIA,LPZ}}$ is reduced from $\dot{h}_{\mathrm{GIA}}$. The debiased GIA-induced BEC is

$$\tilde{\dot{h}}_{\mathrm{GIA}} = \dot{h}_{\mathrm{GIA}} - \bar{\dot{h}}_{\mathrm{GIA,LPZ}}. \tag{12}$$

From which we derive the debiased apparent GIA-mass trend

$$\tilde{\dot{m}}_{\mathrm{GIA}} = \tilde{\dot{h}}_{\mathrm{GIA}} \cdot \rho_{\mathrm{GIA}}. \tag{13}$$

Second, the *LPZ-based GRACE bias correction* $\bar{\dot{m}}_{\mathrm{grav,LPZ}}$ is applied. Prior to determining the mass-balance, a bias correction is applied to the total-mass change derived from time-variable gravity fields. ADC from gravimetry are calibrated to the LPZ by removing the mean ADC in this area $\bar{\dot{m}}_{\mathrm{grav,LPZ}}$. The debiased gravimetric ADC is

$$\tilde{\dot{m}}_{\mathrm{grav}} = \dot{m}_{\mathrm{grav}} - \bar{\dot{m}}_{\mathrm{grav,LPZ}}. \tag{14}$$



The debiased ice-mass trend is

$$\dot{m}_{\mathrm{ice}} = \dot{m}_{\mathrm{grav}} - \dot{m}_{\mathrm{GIA}}. \tag{15}$$

Note that the gravimetric bias correction is not applied to $\dot{m}_{\mathrm{grav}}$ used in the initial combination (Eq. 9). We investigate the four options that arise from either biased or debiased GIA-induced BEC and either biased or debiased total-mass change.

## 2.3 Filtering

A consistent spatial resolution of the data and models is required for the combination in the spatial domain. Moreover, a further noise suppression of GRACE-derived trends is required (Sect. 3.2). Strictly speaking, only a filtered version of $\dot{m}_{\mathrm{grav}}$ is available, since a de-striping filter is applied ($\mathcal{F}_{\mathrm{DS}}(\dot{m}_{\mathrm{grav}})$). A consistent filtering of the quotient $(\dot{m}_{\mathrm{grav}})/(\rho_{\mathrm{GIA}} - \rho_\alpha)$ is therefore not possible. Pragmatically, components with a similar spatial resolution are combined and can be filtered with a Gaussian filter $\mathcal{F}$ afterwards. Hence, we obtain a filtered GIA-induced BEC

$$\tilde{\mathcal{F}}(\dot{h}_{\mathrm{GIA}}) = \frac{\mathcal{F}(\mathcal{F}_{\mathrm{DS}}(\dot{m}_{\mathrm{grav}}))}{\mathcal{F}(\rho_{\mathrm{GIA}} - \rho_\alpha)} - \mathcal{F}\left( \frac{\rho_\alpha(\dot{h}_{\mathrm{alt}} - \dot{h}_{\mathrm{firn}}) - \dot{m}_{\mathrm{firn}}}{\rho_{\mathrm{GIA}} - \rho_\alpha} \right). \tag{16}$$

For integrating mass trends in space, the signal redistribution (leakage) is taken into account by a buffer zone equal to the half-response width of the Gaussian filter appended to the grounding line of the ice sheet (Sect. 4.2). We do not correct for leakage through ocean mass signal separately as it amounts to only 4.5 Gt a$^{-1}$ (Gunter et al., 2014).

## 2.4 Uncertainty characterisation of firn process models

In Equation (9), assumptions on uncertainties of the FDM and altimetry are crucial. In Gunter et al. (2014), $\sigma_{\dot{h}_{\mathrm{alt}}}$ is taken from the formal uncertainty of the least-squares estimation. $\sigma_{\dot{h}_{\mathrm{firn}}}$ can be derived in the same way from the estimated trend of FDM SEC for the observation period. Note that both uncertainties are derived from stochastic information of the least-squares estimation rather than from an uncertainty characterisation of the measurements and the model. Beside those *a priori* uncertainties, Gunter et al. (2014) have performed an uncertainty analysis of the combination result. Their SMB-related uncertainty used for

this purpose is set to 10 % of the estimated trend value referring to Rignot et al. (2008). Note that the uncertainty assessment by Rignot et al. (2008), which amounts to 10–30% of the signal, applied to a different physical quantity than $\dot{h}_{\mathrm{firn}}$: namely to the snow accumulation in a drainage basin.

To quantify the error of firn process models, statistics on differences between two models are evaluated. We use differences of trends of cumulated surface mass balance anomalies (cSMBA) and of firn-thickness trends. We assume those differences

are due to modelling error.

## 2.5 Combination of time series

Previous studies combining gravimetry and altimetry are based on linear-seasonal deterministic models over certain periods (Riva et al., 2009; Gunter et al., 2014; Martín-Español et al., 2016a; Sasgen et al., 2017; Engels et al., 2018). However, signals



in the firn and ice layer over the Antarctic Ice Sheet (AIS) show inter-annual changes (Horwath et al., 2012; Ligtenberg et al., 2012; Mémin et al., 2015). In theory, combining observations on time-series level will lead to a linear GIA signal. For $T$ months the vector

$$\mathbf{m}_{\mathrm{grav}} = \{m_{\mathrm{grav}}(t=1), ..., m_{\mathrm{grav}}(t=T)\} \tag{17}$$

contains the differences in mass at month $t = 1, ..., T$ with respect to a reference mass distribution. The combination of all time series is

$$\mathbf{h}_{\mathrm{GIA}} = \frac{\mathbf{m}_{\mathrm{grav}} - \rho_{ID}(\mathbf{h}_{\mathrm{alt}} - \mathbf{h}_{\mathrm{firn}}) - \mathbf{m}_{\mathrm{firn}}}{\rho_{\mathrm{GIA}} - \rho_{ID}}. \tag{18}$$

This requires all data is available as monthly gridded products. To simplify, we assume effective densities do not change over time. To be consistent with the combination on trend level, $\rho_{\mathrm{ID}}$ is replaced with $\rho_\alpha$ from the trend-based approach.

The data and models of every month are filtered similarly to the trend-based approach to make the resolution consistent (Sect. 2.3). Afterwards they are combined according to Eq. 18 which results in a GIA time series for each grid cell.

By assumption the resulting time series $\mathbf{h}_{\mathrm{GIA}}$ include GIA as an approximately linear signal in short time periods (tens of years), e.g. during satellite observation periods (e.g. Huybrechts and Le Meur, 1999). An adjusted trend to $\mathbf{h}_{\mathrm{GIA}}$ will lead to $\dot{h}_{\mathrm{GIA}}$. We are aware that for regions with a low-viscosity asthenosphere, e.g. Pine Island Bay, the linear response is under debate even for decadal periods (Barletta et al., 2018).

## 2.6 Sensitivity analysis

The sensitivity analysis allows for the quantification of the dependency of inverse GIA estimates to different data, models and assumptions. Starting from a reference experiment, certain parameters are changed. Every experiment is performed with and without the two LPZ-based bias corrections to demonstrate their effect. It is examined how different altimetry data (Sect. 3.1), degree-1 and $C_{20}$ products (Sect. 3.2), and the empirically determined errors of the firn-process models (Sect. 4.1) affect the GIA solution. Analogous to Riva et al. (2009) and Gunter et al. (2014) a Gaussian filter (half-response width = 400 km) is applied. For the integration of mass trends over the AIS, the West Antarctic Ice Sheet (WAIS) and the East Antarctic Ice Sheet (EAIS), we also use a buffer zone of 400 km grounding line distance to mitigate leakage. The Antarctic Peninsula (AP) is not considered separately here.

Beside integrated mass trends, a root mean square (RMS) difference of each inverse GIA solution with respect to the reference experiment is calculated, hereinafter referred to as *RMS difference from reference experiment* (RMS$_{\mathrm{RE}}$).

$$\mathrm{RMS}_{\mathrm{RE}} = \sqrt{\frac{1}{N} \sum_{i=1}^{N} \left( \dot{h}_{\mathrm{GIA,comp},i} - \dot{h}_{\mathrm{GIA,ref},i} \right)^2}. \tag{19}$$

Here, $N$ is the number grid cells of a cartesian grid in the polar stereographic projection of the AIS area (EPSG: 3031) including the buffer zone. $\dot{h}_{\mathrm{GIA,comp}}$ refers to the GIA solution which is compared to the reference experiment ($\dot{h}_{\mathrm{GIA,ref}}$). We use $\dot{h}_{\mathrm{RMS}_{\mathrm{RE}}}$ in addition to comparing integrated mass trends because integrated mass trends values may hide regional differences.





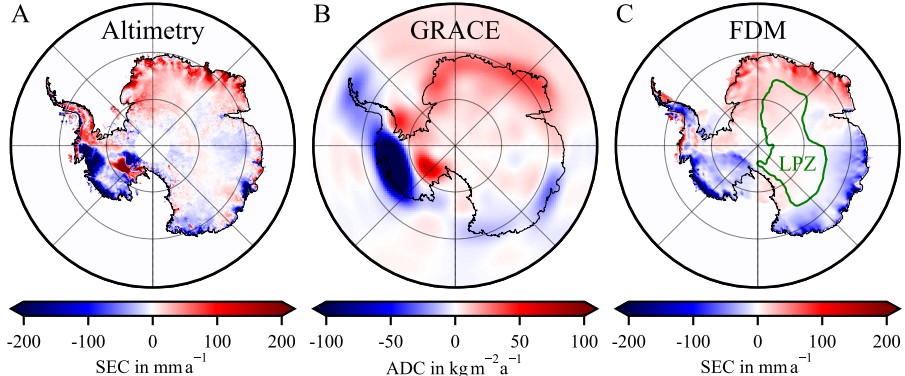

**Figure 1.** A: Altimetry-derived surface elevation change (SEC), B: GRACE-derived area-density changes (ADC), and C: FDM-derived SEC (time period: 2002-04/2016-08). A Gaussian filter was applied to the GRACE result (half-response 250 km). Low precipitation zone (LPZ) (green, C).

The sensitivity to the choice of firn-process models is investigated as follows: Based on the comparison of two firn-process models, empirical samples of error patterns are generated. They are added to $\dot{h}_{\mathrm{firn}}$ and $\dot{m}_{\mathrm{firn}}$ and propagated to the empirical GIA estimates. Additionally, all identified trend differences of cSMBA are added to $\dot{h}_{\mathrm{firn}}$ and $\dot{m}_{\mathrm{firn}}$.

Furthermore, the dependency on differing time periods is investigated. Under the assumption that GIA is linear in time, the used time interval should have negligible influence. While the time interval for the reference experiment is 2003-03/2009-10 (according to Gunter et al. (2014)), alternative periods are the main GRACE observation period (2002-04/2016-08) and the overlap period between GRACE and CryoSat-2 (2010-07/2016-08).

## 3 Data and models

This section specifies the data sets and processing steps used in the sensitivity experiments. The information is summarised in Table 1. Furthermore, models and assumptions for further elaboration are explained. Reference system parameters are chosen according to the IERS Conventions (Petit and Luzum, 2010).

### 3.1 Altimetry

The SEC from the Multi-Mission altimetry (MM-Altimetry) from Schröder et al. (2019) is estimated by a repeat-altimetry approach. The data from the missions Seasat, Geosat, ERS-1, ERS-2, Envisat, ICESat and CryoSat-2 are combined resulting in a monthly sampled time series on a 10 km grid. The reader is referred to Schröder et al. (2019) for details on processing and background information. In order to combine the time series with GRACE, we use the monthly results from 2002-04 at the earliest to 2016-08 at the latest which involves observations of the missions ERS-2, Envisat, ICESat and CryoSat-2 (Fig. 1A). This is because we use GRACE monthly solutions during this time period (Fig. 1B). However, the altimetry missions have a different spatial and temporal sampling. For this reason every month of the combined time series differs in spatial coverage.





We obtain a linear rate over the respective intervals by adjusting an offset and a linear trend to the MM time series for each cell of the 10 km grid. For the reference experiment no annual-periodic signal is co-estimated in order to be consistent with Gunter et al. (2014). We apply weights according to the uncertainty estimates of each epoch of the MM time series. We took the criterion that the trend would only be estimated for a grid cell if more than five observation months are available, and at least 80 % of the selected total time span is covered. This criterion should avoid outlier trends through insufficient sampling.

The uncertainty $\sigma_{\dot{h}_{\mathrm{alt}}}$ used in Eq. 11 is the *a posteriori* standard deviation derived from the least-squares adjustment of the MM time series.

To investigate how the choice of altimetry products affects the GIA estimation, single-mission time series are calculated for Envisat and ICESat. They consistently use the same processing steps as the MM altimetry from Schröder et al. (2019), with the exception that the final step of weighted spatio-temporal smoothing is applied to single-mission data rather than multi-mission

data. In total three different altimetry time series are used for testing the gravimetry-altimetry combination approach. To assess the sensitivity of results to the co-estimation of seasonal signals, an additional version of the MM altimetry trends is calculated by co-estimating the annual sinusoidal signal (*MM seasonal* in Table 1). This is consistent with the treatment of GRACE and the firn-process models.

Part of the altimetry-derived SEC is caused by the elastic BEC of the solid Earth by present-day ice-mass change ($\dot{h}_{\mathrm{elastic}}$).

This is taken into account by scaling $\dot{\tilde{h}}_{\mathrm{alt}}$ by a factor of 1.015 (Riva et al., 2009). The influence on the GIA estimate is negligible (Gunter et al., 2014).

## 3.2 Gravimetry

GRACE-derived monthly mass variations are calculated from the ITSG-Grace2016 monthly gravity field solutions up to degree and order 90 (Mayer-Gürr et al., 2016) using Eq. (1). Monthly solutions from other processing centres are not considered be-

cause ITSG-Grace2016 is identified through internal comparison as the gravity field solution series with highest signal-to-noise ratio. The influence of the different GRACE monthly solutions on the inverse GIA result was shown and discussed in Gunter et al. (2014). We do not use solutions after 2016-08. Those solutions show a much higher noise level due to accelerometer issues.

GRACE monthly solutions need to be complemented by the degree-1 term of the spherical harmonic coefficients, as this is

not observed by GRACE. Three different products to replace the degree-1 coefficients are evaluated: (1) A product is determined following Swenson et al. (2008) using ITSG-Grace2016 monthly solutions (*d1_ITSG*). (2) A Satellite Laser Ranging (SLR) product by Cheng et al. (2013b) (*d1_SLR*) and (3) degree-1 coefficients by Rietbroek et al. (2016) are used (*d1_ITG*).

Furthermore, the influence of the flattening term $C_{20}$ is investigated. It is replaced by external products because this coefficient is only poorly determined by GRACE (Cheng and Ries, 2017). Three different products are compared: (1) SLR based

time series are used from the Center for Space Research at University of Texas, USA (*c20_SLR_CSR*, Cheng et al. (2013a)); (2) SLR based time series from the German Research Centre for Geosciences, Potsdam, Germany (*c20_SLR_GFZ*, König et al. (2019)); (3) and a time series from the Delft University of Technology, Delft, Netherlands (*c20_TU_Delft*), which is derived from GRACE observations themselves and an ocean model (Sun et al., 2015).



A critical point is filtering because the monthly solutions are noisy and have a correlated error pattern (Horwath and Dietrich, 2009). A destriping-filter is applied in the spherical-harmonic domain (Swenson and Wahr, 2006).

A linear-seasonal model is adjusted to the filtered Stokes coefficients (offset, linear, annual-periodic and 161-day periodic). The trend is synthesised from the spherical-harmonic into the spatial domain on the altimetry grid with 50 km resolution. In this way for each grid cell a linear area-density trend in $\mathrm{kg\,m^{-2}\,a^{-1}}$ is determined (Fig. 1B).

## 3.3   Firn-process models

As shown in the combination approach (Eq. 10), information on density variations of the firn layer is required. SMB is the sum of precipitation, snow drift, sublimation and meltwater runoff. The SMB components are numerically simulated with the RACMO2.3p2 model containing a multi-layer snow model developed by the Royal Netherlands Meteorological Institute (KNMI) and the Institute for Marine and Atmospheric Research, Utrecht, Netherlands (IMAU) (van Wessem et al., 2018).

These results are compared to the MAR model of the Laboratory of Climatology, Liège, Belgium (Agosta et al., 2019). The regional climate models are forced at its lateral boundaries with the ERA-40 and ERA Interim reanalyses. Mass fluxes (snowfall, snow drift, sublimation, erosion/deposition, and surface melt) as well as surface temperature are then used to force an off-line firn densification model that includes firn compaction, vertical meltwater transport and refreezing, and thermodynamics of the firn layer.

The RACMO2 and MAR SMB product are appropriate for comparison as both are similar in terms of temporal (monthly) and spatial resolution (RACMO2: 27 km, MAR: 35 km). Moreover, both variants considered here use the same forcing. There is no independent knowledge (in a spatial resolution similar to that of SMB models) about the ice flow contribution to ice mass balance, and hence about the degree of balance or imbalance between SMB and ice flow. Therefore, the modelled SMB is only used to derive SMB-induced mass variations with respect to any background signal of mass change. The unknown

background signal of mass change is the possible imbalance between the mean SMB over a multi-year reference period and the mean effect of ice flow on mass balance over the the same reference period. The considered SMB-induced mass variations hence arise from the temporal cumulation of SMB anomalies with respect to the mean SMB over the reference period. Here, we define the reference period to be the entire model period for RACMO2.3p2 and MAR (1979-01/2016-12). For the satellite observation periods (e.g. 2002-04/2016-08) the surface mass trend ($\dot{m}_{\mathrm{firn}}$) or literally, the trend of cumulated surface mass

balance anomalies (cSMBA) is estimated (co-estimated with bias and annual-periodic).

The used firn model has also been developed at IMAU (Ligtenberg et al., 2011) and is called IMAU-FDM. It is forced at the upper boundary by SMB components from RACMO2 (precipitation, sublimation, erosion, melt) and internally calculates densification and refreezing. In IMAU-FDM, the firn layer is initialized by forcing it repeatedly with the 1979-2016 surface mass fluxes and temperature, until an equilibrium firn layer is established. It implies that, in the model, present-day conditions

represent a state of equilibrium and that there is no net firn thickness change over the model period 1979-01/2016-12. One result of the actual model run is the firn-elevation-change time series. A linear-seasonal model (bias, trend, annual-periodic) of firn-process-induced SEC is adjusted to the FDM time series for the observation periods under investigation (Fig. 1C).



The LPZ (Fig. 1C) is defined based on ECMWF ERA-Interim reanalysis precipitation product. We use $20\,\mathrm{mm\,a^{-1}}$ annual precipitation as a threshold for low precipitation (rather than $21.9\,\mathrm{mm\,a^{-1}}$ used by Gunter et al. (2014)).

The trend-differences between RACMO2.3p2 and MAR SMB products are used for uncertainty characterisation of firn process models. In order to gain statistical information on possible trend differences over a 7-year interval, we calculate trend differences over 32 intervals of 7 years length (1979-01/1965-12; 1980-01/1966-12; ... ; 2010-1/2016-12) covered by RACMO2.3p2 and MAR. The 7-years length is the approximate length of the observation period of our reference inverse experiment (2003-03/2009-10) defined by the ICESat observation period. An FDM forced with MAR SMB does not exist. However, the RACMO2.3p2 SMB and the derived FDM are directly linked to each other. For this reason we assume that derived conclusions on errors of SMB are transferable to the FDM as a lower bound. Pseudo FDM-trend differences are estimated out of the cSMBA trends by

$$\Delta \dot{h}_{\mathrm{firn},j} = \frac{\Delta \dot{m}_{\mathrm{firn},j}}{\rho_{\mathrm{MAR}}}. \tag{20}$$

$\Delta \dot{m}_{\mathrm{firn},j}$ is the $j$-th trend difference between cSMBA. $\rho_{\mathrm{MAR}}$ is calculated from MAR density fields by taking their average over the near-surface layers (0–1 m) and over the whole model period. This does not consider the correct evolution of the firn layer by MAR model results. Furthermore, uncertainties through equilibrium assumptions are still not considered and need further investigation.

Prior to the combination, cSMBA and FDM trends are linearly interpolated to the polar-stereographic grid. The high-resolution products (altimetry and firn-process models) are modified as follows: NaN-Grid cells on the grounded part of the ice sheet (missing data) are treated as case 3 in Eq. (10).

### 3.4 Density assumptions

The ratio between volume and area-density changes of the superimposed processes GIA, firn variations and ice dynamics is described by the effective densities $\rho_{\mathrm{GIA}}$, $\rho_{\mathrm{firn}}$ and $\rho_{\mathrm{ID}}$. The latter is assumed to be $917\,\mathrm{kg\,m^{-3}}$. A general statement is not possible for variations in the firn layer, as the firn density is variable in space and time. The location-dependent estimation for $\rho_{\mathrm{firn}}$ is calculated using the empirical Eq. (2) in Ligtenberg et al. (2011).

The density mask for $\rho_{\mathrm{GIA}}$ is generated as follows: The ratio between the GIA-induced BEC and the GIA-induced ADC change is about $3700\,\mathrm{kg\,m^{-3}}$ (Wahr et al., 2000). We use $4000\,\mathrm{kg\,m^{-3}}$ over the Antarctic continent and $3400\,\mathrm{kg\,m^{-3}}$ under the ice-shelves and the ocean with a smooth transition (according to Riva et al. (2009); Gunter et al. (2014)). These numbers account for the redistribution of ocean mass through GIA and are derived from forward-model results. This density is not a density in a material-science sense. It is an effective value which sets GIA-induced BEC and the ADC in relation. The term *rock* used in literature might be misleading.





**Table 1.** Overview of all performed experiments of sensitivity analysis (Sect. 2.6 and 4.2, Table 2). All experiments use ITSG-Grace2016 monthly solutions (Mayer-Gürr et al., 2016) over 2003-03/2009-10 time period, except for the last two experiments which use the quoted time period.

| Experiment | Degree-1 repl. Section 3.2 | $C_{20}$ repl. Section 3.2 | used Altimetry Section 3.1 | used firn-process model Section 3.3 |
|---|---|---|---|---|
| *reference* | d1_ITSG | c20_SLR_CSR | Multi-Mission (incl. ERS-2, Envisat, ICESat) | RACMO2.3p2 |
| *d1_SLR* | d1_SLR | c20_SLR_CSR | Multi-Mission | RACMO2.3p2 |
| *d1_ITG* | d1_ITG | c20_SLR_CSR | Multi-Mission | RACMO2.3p2 |
| *c20_SLR_GFZ* | d1_ITSG | c20_SLR_GFZ | Multi-Mission | RACMO2.3p2 |
| *c20_TU_Delft* | d1_ITSG | c20_TU_Delft | Multi-Mission | RACMO2.3p2 |
| *ICESat-only* | d1_ITSG | c20_SLR_CSR | ICESat | RACMO2.3p2 |
| *Envisat-only* | d1_ITSG | c20_SLR_CSR | Envisat | RACMO2.3p2 |
| *MM seasonal* | d1_ITSG | c20_SLR_CSR | Multi-Mission, co-estimation of seasonal components | RACMO2.3p2 |
| *RACMO2+EOFx* | d1_ITSG | c20_SLR_CSR | Multi-Mission | RACMO2.3p2 with empirical orthogonal functions (EOF) of firn-process uncertainty (Section 4.1) |
| *2010-07/2016-08* | d1_ITSG | c20_SLR_CSR | Multi-Mission (incl. Envisat, CryoSat-2) | RACMO2.3p2 |
| *2002-04/2016-08* | d1_ITSG | c20_SLR_CSR | Multi-Mission (incl. ERS-2, Envisat, ICESat, CryoSat-2) | RACMO2.3p2 |

## 4 Results

### 4.1 SMB uncertainty

There are considerable differences between the time series of cSMBA from the RACMO2 and MAR SMB product for each cell. Figure 2 shows the integrated values for the AIS. We use the 32 trend differences from the moving 7-year-intervals to quantify discrepancies of derived cSMBA trends between both models. Figure 3 shows the RMS of all trend differences and compares it

5   with the formal uncertainty from the least-squares estimation and with the 10 % uncertainty assumption (Sect. 2.4). The latter two are derived from the $\dot{m}_{\text{firn}}$ of the RACMO2.3p2 SMB product over the ICESat-observation period (2003-03/2009-10). The formal uncertainty and the 10 % assumption are similar in spatial pattern and magnitude. The standard deviation of trend differences is similar in spatial pattern, too, but approximately three times larger in magnitude.

To extract the dominant error patterns, a spectral decomposition of the 32 7-year trend differences (cf. Sect. 3.3) is done

10  by a principal-component analysis (using singular value decomposition). Hence, the dominant empirical orthogonal functions (EOF) and accompanying principal components are computed. From this analysis we obtain the dominant error patterns that are uncorrelated to each other and capture characteristic features of uncertainty. The first three EOFs of the trend differences explain ~68 % of the total variance (Fig. 4A–C). Figure 4D shows the principle components indicating the scaling of corresponding EOF. For instance, EOF-1 is dominated by variations in WAIS. EOF-2 shows more variations on smaller scales. Without an

15  attempt to further interpret the patterns of trend differences between the two models, the explored trend differences are used here to investigate the sensitivity of the inverse GIA estimates to these differences characterising firn process uncertainty. For this purpose, (1) we add the EOFs to the firn process trends ($\dot{m}_{\text{firn}}$, $\dot{h}_{\text{firn}}$), which we use as input for the data combination. From



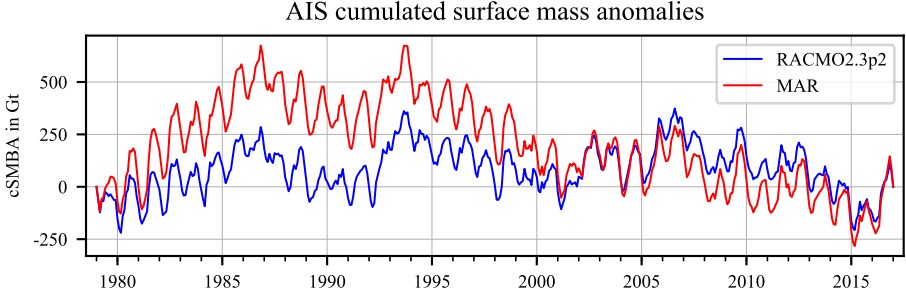

**Figure 2.** Cumulated surface mass balance anomalies (cSMBA) of the regional climate models RACMO2.3p2 (blue, van Wessem et al. (2018)) and MAR (red, Agosta et al. (2019)), integrated over the grounded AIS.

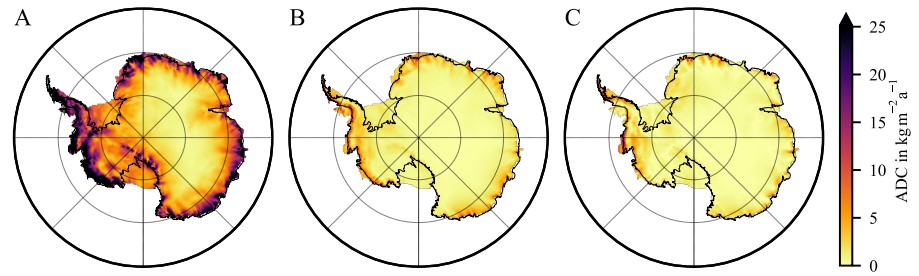

**Figure 3.** Three uncertainty assessments for the area density change (ADC) trend induced by cumulated surface mass balance anomalies (cSMBA). A: RMS of cSMBA trend differences between RACMO2.3p2 and MAR for all 7-year intervals (Sect. 3.3), B: the formal uncertainty from least-squares estimation for 2003-03/2009-10, and C: the 10 % uncertainty assumption.

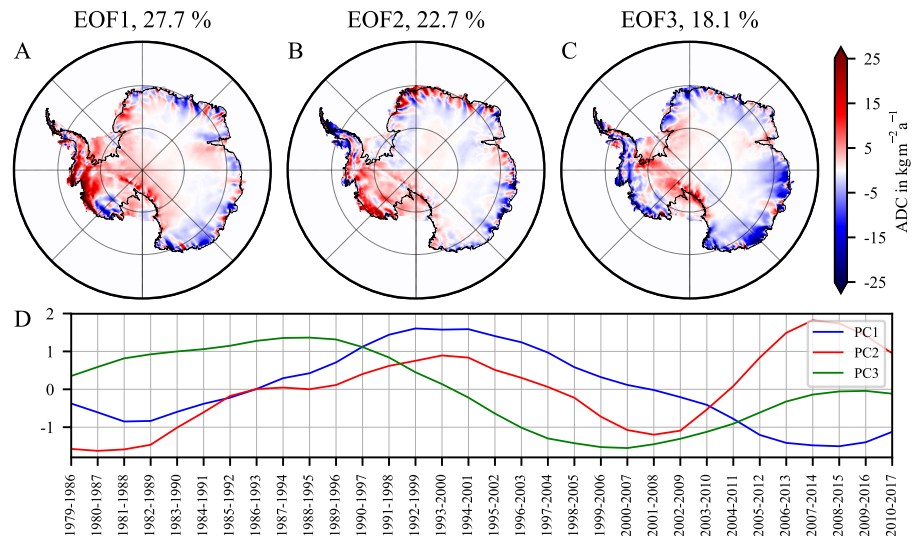

**Figure 4.** A–C: Area-density change (ADC) of the first three EOFs of the trend differences between RACMO2.3p2 and MAR cumulated surface mass balance anomalies (cSMBA). D: the respective principal components (PC).



**Table 2.** Results from the sensitivity experiments. This table is structured like Table 2 in Gunter et al. (2014). Each line reports results from one experiment, where line one reports the reference experiment. The time period is 2003-03/2009-10 except where it is quoted by experiment name. Column 1: experiment name, according to Table 1. Column 2: RMS difference of the GIA-induced bedrock elevation change (BEC) estimate ($RMS_{RE}$) to the reference experiment. Columns 3 and 4: applied LPZ-based bias correction (cf. Section 2.2) for GIA-induced BEC and GRACE area-density change, respectively. Columns 5, 6, 7: spatial integral of total-mass change (Eq. 14) over the Antarctic Ice Sheet (AIS), the West Antarctic Ice Sheet (AIS) and the East Antarctic Ice Sheet (EAIS), including a 400 km buffer zone. Columns 8–10 and 11–13: Same as column 5–7, but for the apparent GIA-mass change (Eq. 13) and for the ice-mass change (Eq. 15), respectively. Numbers in brackets give results of experiments with no bias corrections.

| Experiment | $RMS_{RE}$ | LPZ bias | | Total-mass change | | | apparent GIA-mass change | | | Ice-mass change | | |
|---|---|---|---|---|---|---|---|---|---|---|---|---|
| | | GIA | GRACE | AIS | WAIS | EAIS | AIS | WAIS | EAIS | AIS | WAIS | EAIS |
| | $mm\,a^{-1}$ | $mm\,a^{-1}$ | $kg\,m^{-2}\,a^{-1}$ | $Gt\,a^{-1}$ | | | $Gt\,a^{-1}$ | | | $Gt\,a^{-1}$ | | |
| *reference* | 0.0 | 1.6 | 1.9 | -40 | -78 | 39 | 44 | 21 | 24 | -84 | -99 | 15 |
| | (1.6) | (0.0) | (0.0) | (0) | (-68) | (68) | (172) | (53) | (119) | (-173) | (-121) | (-51) |
| **degree-1** | | | | | | | | | | | | |
| *d1_SLR* | 0.1 | 2.0 | 3.2 | -42 | -79 | 38 | 43 | 20 | 23 | -85 | -99 | 15 |
| | (2.0) | (0.0) | (0.0) | (25) | (-62) | (86) | (199) | (60) | (139) | (-174) | (-122) | (-53) |
| *d1_ITG* | 0.1 | 1.8 | 2.5 | -41 | -80 | 39 | 43 | 19 | 24 | -84 | -99 | 15 |
| | (1.8) | (0.0) | (0.0) | (12) | (-66) | (78) | (185) | (55) | (130) | (-173) | (-121) | (-52) |
| **$C_{20}$** | | | | | | | | | | | | |
| *c20_SLR_GFZ* | 0.0 | 1.4 | 1.2 | -39 | -78 | 39 | 46 | 21 | 25 | -85 | -99 | 15 |
| | (1.4) | (0.0) | (0.0) | (-14) | (-72) | (57) | (157) | (49) | (108) | (-171) | (-121) | (-50) |
| *c20_TU_Delft* | 0.1 | 1.0 | -0.4 | -36 | -77 | 42 | 48 | 21 | 26 | -83 | -99 | 15 |
| | (1.1) | (0.0) | (0.0) | (-43) | (-79) | (36) | (127) | (41) | (85) | (-170) | (-121) | (-49) |
| **Altimetry** | | | | | | | | | | | | |
| *ICESat-only* | 1.1 | 1.1 | 1.9 | -40 | -78 | 39 | 59 | 20 | 39 | -99 | -98 | -1 |
| | (1.7) | (0.0) | (0.0) | (0) | (-68) | (68) | (142) | (41) | (101) | (-142) | (-109) | (-34) |
| *Envisat-only* | 0.8 | 1.5 | 1.9 | -40 | -78 | 39 | 54 | 33 | 22 | -94 | -111 | 17 |
| | (1.8) | (0.0) | (0.0) | (0) | (-68) | (68) | (174) | (63) | (111) | (-174) | (-131) | (-43) |
| *MM seasonal* co-estimated | 0.1 | 1.7 | 1.9 | -40 | -78 | 39 | 46 | 21 | 25 | -86 | -99 | 14 |
| | (1.7) | (0.0) | (0.0) | (0) | (-68) | (68) | (177) | (54) | (122) | (-177) | (-122) | (-55) |
| **Firn-process error** | | | | | | | | | | | | |
| *RACMO2+EOF1* | 0.5 | 1.8 | 1.9 | -40 | -78 | 39 | 48 | 29 | 18 | -87 | -108 | 20 |
| | (1.9) | (0.0) | (0.0) | (0) | (-68) | (68) | (190) | (65) | (124) | (-190) | (-133) | (-57) |
| *RACMO2+EOF2* | 0.3 | 1.7 | 1.9 | -40 | -78 | 39 | 51 | 31 | 20 | -90 | -109 | 19 |
| | (1.8) | (0.0) | (0.0) | (0) | (-68) | (68) | (181) | (64) | (117) | (-181) | (-132) | (-50) |
| *RACMO2+EOF3* | 0.3 | 1.6 | 1.9 | -40 | -78 | 39 | 41 | 20 | 21 | -80 | -98 | 18 |
| | (1.6) | (0.0) | (0.0) | (0) | (-68) | (68) | (169) | (52) | (117) | (-169) | (-120) | (-49) |
| **Time interval** | | | | | | | | | | | | |
| *2002-04/2016-08* | 1.1 | 1.8 | 3.5 | -121 | -160 | 39 | 18 | -4 | 22 | -140 | -156 | 17 |
| | (1.7) | (0.0) | (0.0) | (-48) | (-141) | (93) | (158) | (32) | (126) | (-205) | (-172) | (-33) |
| *2010-07/2016-08* | 1.4 | 2.2 | 5.3 | -181 | -189 | 8 | 67 | 37 | 30 | -248 | -227 | -21 |
| | (2.9) | (0.0) | (0.0) | (-70) | (-160) | (90) | (239) | (81) | (158) | (-309) | (-241) | (-68) |
| **Combination on time-series level** | | | | | | | | | | | | |
| *2010-07/2016-08* | | 2.1 | 5.3 | -181 | -189 | 8 | 39 | 17 | 23 | -220 | -206 | -14 |
| | | (0.0) | (0.0) | (-70) | (-160) | (90) | (207) | (59) | (148) | (-277) | (-219) | (58) |





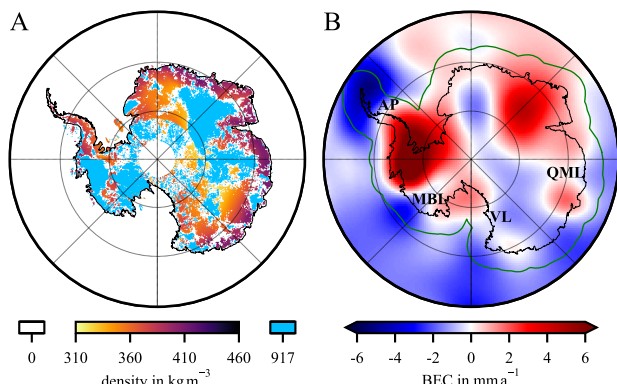

**Figure 5.** A: Estimated $\rho_\alpha$-density (Eq. 10) of reference experiment. B: GIA-induced bedrock elevation change (BEC) of the reference experiment (RMS: 2.2 mm a$^{-1}$), 400 km buffer zone (green line), geographical regions indicated: Antarctic Peninsula (AP), Marie Byrd Land (MBL), Victoria Land (VL), Queen Mary Land (QML). For results from the other simulation experiments see Figure S4 and S5.

this we get three GIA estimates to be compared with our reference solution. (2) We add each trend difference separately to the firn process trends resulting in another 32 GIA estimates.

### 4.2 Sensitivity analysis

Inverse GIA estimates are calculated using different choices of: (1) degree-1 solutions, (2) $C_{20}$ substitutions, (3) altimetry products, (4) empirical orthogonal functions (EOF) of firn-process errors and (5) time intervals (Table 1). The reference experiment

refers to the time period 2003-03/2009-10 and uses MM-Altimetry-derived SEC, ITSG-Grace2016 monthly solution (degree-1: d1_ITSG, C20: SLR_CSR), and the firn-process trends from RACMO2.3p2 over this period. The RMS of the reference GIA-induced BEC estimate is 2.2 mm a$^{-1}$. The estimated $\rho_\alpha$ (Eq. 10) is shown in Fig. 5A. Apart from the gridded GIA-induced BEC (Fig. 5B, S5), we compare the integrated trends $\dot{\tilde{m}}_{\text{grav}}$, $\dot{\tilde{m}}_{\text{GIA}}$, and $\dot{\tilde{m}}_{\text{ice}}$ leading to *total-mass change* (from GRACE), apparent *GIA-mass change*, and *ice-mass change*, respectively. The results are summarised in Table 2. Furthermore, the RMS$_{\text{RE}}$

(Eq. 19) quantifies the discrepancy to the reference experiment GIA estimate. Figure 6 shows the mass-balance estimates for 2003-03/2009-10.

Biased total mass changes for different $C_{20}$ and degree-1 products vary between -43 Gt a$^{-1}$ (c20_TU_Delft) and +25 Gt a$^{-1}$ (d1_SLR), that is in a range of 68 Gt a$^{-1}$. Debiased total-mass change (Eq. 14) only differ by 6 Gt a$^{-1}$ for the same time period (Table 2). In Figure 6 biased and debiased total-mass changes of the entire AIS are illustrated. Note that biased total-mass

change of 0 Gt a$^{-1}$ in Table 2 arises coincidentally by used input data.

The biased apparent GIA-mass change of the AIS with MM-Altimetry (reference experiment) is very close to the Envisat-only estimate (174 vs. 172 Gt a$^{-1}$). The biased ICESat-only result differs from the reference experiment by about 30 Gt a$^{-1}$ (142 vs. 172 Gt a$^{-1}$). Debiased estimates that use Envisat-only or ICESat-only results differ from estimate of the reference experiment by 10 and 15 Gt a$^{-1}$, respectively. The differences due to the co-estimation of seasonal components are marginal (~2 Gt a$^{-1}$).





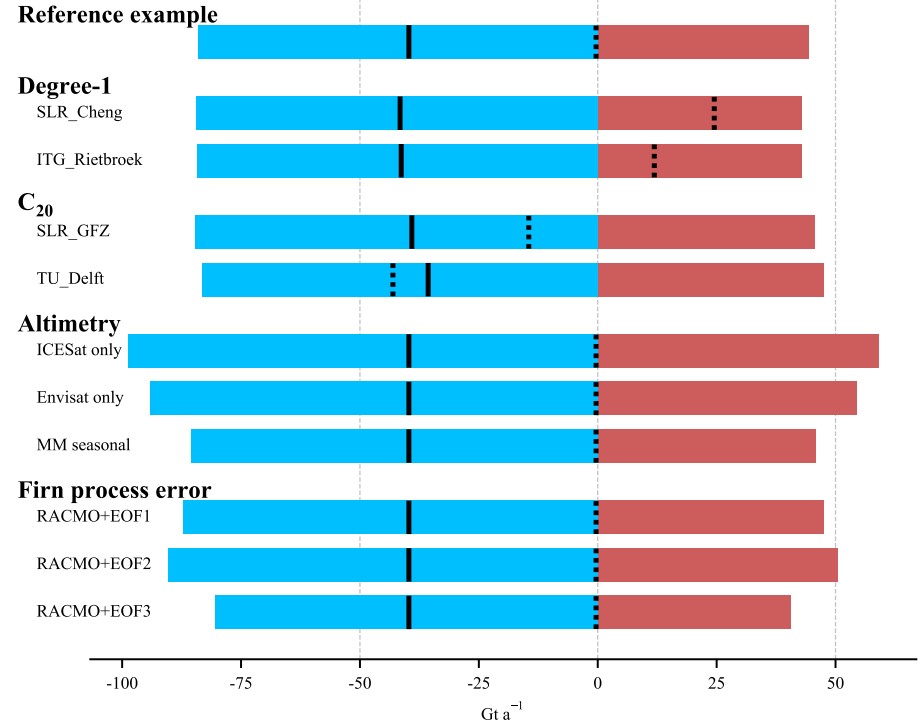

**Figure 6.** Mass change results for the entire AIS over the interval 2003-03/2009-10 from experiments with different data products and methodological choices. The LPZ-based bias correction was applied. Debiased total-mass change (solid black lines) is separated into debiased GIA-mass (red) and ice-mass change (blue). Dotted lines show the total mass changes that arise when no bias corrections are applied. The case of no bias correction is further illustrated in Fig. S7.

Applying the approach to different time intervals 2002-04/2016-08 and 2010-07/2016-08 leads to debiased total-mass changes of -121 and -181 Gt a$^{-1}$, respectively (biased estimates: -48 and -70 Gt a$^{-1}$).

    The addition of the determined EOFs (Sect. 4.1) propagates to differences of the GIA solution of up to 7 Gt a$^{-1}$ for the debiased GIA-mass change and up to 18 Gt a$^{-1}$ for the biased GIA-mass change. Additionally, Figure S6 shows the standard deviation of the 32 GIA estimates resulting from propagating the 32 trend differences between RACMO2 and MAR.

5   **4.3   Combination on time-series level**

Gravimetry, altimetry, SMB and FDM are available as monthly gridded products with sufficient spatial coverage from 2010-07 to 2016-08 due to the availability of GRACE, CryoSat-2 and RACMO2.3p2. Riva et al. (2009) and Gunter et al. (2014) only use ICESat altimetry data, which does not allow a monthly sampling, as it has only 2–3 monthly observation intervals per year.

    We used the estimated $\rho_\alpha$ from the trend-based combination during the same time interval (Fig. S4I) to be consistent for 10   comparison. Figure 7 shows the GIA-induced mass-change time series for AIS (with 400 km buffer-zone). For applying the

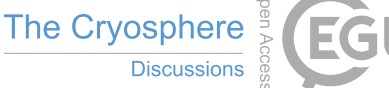

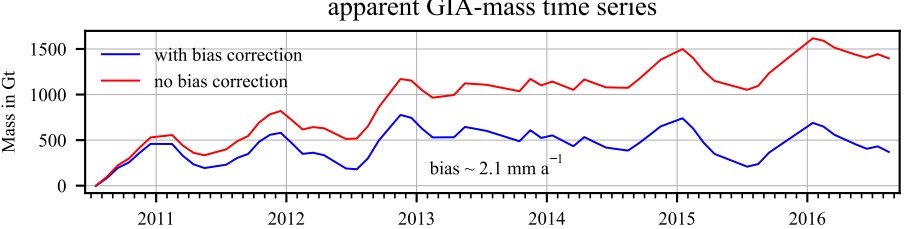

**Figure 7.** The apparent GIA-mass time series of the AIS (with 400 km buffer zone) resulting from the combination of the monthly gridded time series (2010-07/2016-08) with (blue) and without (red) LPZ-based bias correction of the determined GIA signal.

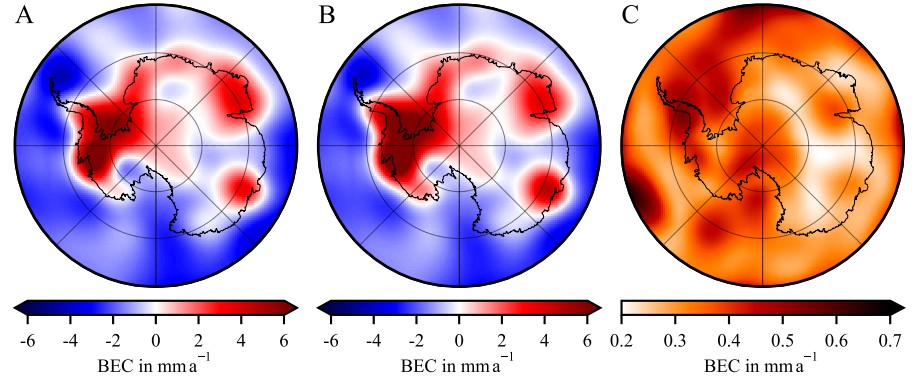

**Figure 8.** For 2010-07/2016-08 time period. A: Debiased GIA bedrock elevation change (BEC) by combining time series of all data sets and models, B: combination of trends, and C: the formal uncertainty from least-squares estimation.

LPZ-based GIA bias correction, the linear GIA trend in the LPZ is estimated (offset and trend only). Figure 8A shows the debiased GIA-induced BEC based on the time series combination. Figure 8C shows its formal uncertainty from least-squares estimation, which should be considered as a lower bound. For comparison, Fig. 8B shows the GIA-induced BEC following the trend-based combination approach. The GIA-induced apparent mass changes from the combination on time-series and trend level are 39 and 67 Gt a$^{-1}$ for AIS, 17 and 37 Gt a$^{-1}$ for WAIS, and 23 and 30 Gt a$^{-1}$ for EAIS, respectively (Table 2). The ice-mass changes are -220 and -248 Gt a$^{-1}$ for AIS, -206 and -227 Gt a$^{-1}$ for AIS, and -14 and -21 Gt a$^{-1}$ for EAIS, respectively. The integrated formal uncertainty of the apparent GIA-mass change for AIS with 400 km buffer zone is 25 Gt a$^{-1}$ (Fig. 8C).

## 5 Discussion

Since the aim of this study is to examine the sensitivity of the inverse approach towards several data input and methodological choices, differences to the reference experiment are discussed on the basis of selected processing parameters.



**Table 3.** The comparison of integrated mass changes from combination used in this study and those published in Gunter et al. (2014). For this we used GFZ RL05 GRACE solutions, ICESat-only altimetry, and RACMO2.1 products during 2003-03/2009-10.

| Solution | Total-mass change in Gt a$^{-1}$ | | | apparent GIA-mass change in Gt a$^{-1}$ | | | Ice-mass change in Gt a$^{-1}$ | | |
|---|---|---|---|---|---|---|---|---|---|
| | **AIS** | **WAIS** | **EAIS** | **AIS** | **WAIS** | **EAIS** | **AIS** | **WAIS** | **EAIS** |
| This study | -51 | -90 | 39 | 49 | 12 | 37 | -100 | -102 | 2 |
| Gunter et al. (2014) | -45 | -86 | 41 | 54 | 18 | 36 | -99 | -104 | 5 |

## 5.1 Assessment of the results

We performed a test run with similar input data as used in Gunter et al. (2014) to test our data processing. We used GFZ RL05 GRACE solutions, ICESat Altimetry, and the RACMO2.1 SMB product (and corresponding IMAU-FDM). Table 3 shows the comparison of both results. AIS total-mass, apparent GIA-mass and ice-mass change estimates reproduce results by Gunter et al. (2014) to within 6, 5 and 1 Gt a$^{-1}$, respectively. Those differences might be attributed to a slightly different LPZ and altimetry processing. Gunter et al. (2014) indicate that the uncertainty for the apparent GIA-mass and ice-mass change from various GRACE solutions and filtering variants is 40 Gt a$^{-1}$ and 44 Gt a$^{-1}$, respectively.

In general our GIA estimates (Fig. 5B) shows a similar spatial pattern compared to estimates by Gunter et al. (2014). Nonetheless, especially the AP, Marie Byrd Land (MBL), Victoria Land (VL), and Queen Mary Land (QML) show larger differences.

In the AP, altimetry-derived SEC are available for a part of the area only (Fig. S1). As a result, GRACE-derived area-density changes can be attributed mainly to GIA-mass change, as altimetry is missing. The result is an unphysical, negative GIA BEC. The negative signal in MBL is of a similar order of magnitude as in Riva et al. (2009) and Sasgen et al. (2017). A negative GIA signal in QML can be found in Martín-Español et al. (2016a). The uncertainty of the GIA signal is sometimes so large, that even its sign cannot be determined.

For example, propagating trend differences between RACMO2.3p2 and MAR cSMBA products to GIA estimates (Fig. S6) leads to a high standard deviation of the GIA signal in MBL and Victoria Land (VL). Even forward models show large variations in the spatial pattern of the GIA-induced BEC with a different sign of BEC (Martín-Español et al., 2016b; Whitehouse et al., 2019).

## 5.2 Sensitivity to degree-1 and C$_{20}$-products and the effect of bias estimation

The use of several degree-1 and C$_{20}$-products for the GRACE processing leads to a differing total-mass trend for the AIS (Barletta et al., 2013). In supplementary material of Gunter et al. (2014) the influence of two different degree-1 products has been shown. Here we show how the bias corrections eliminate those differences in total-mass and apparent GIA-mass change (Sect. 4.2, Table 2). The RMS$_{RE}$ of all debiased GIA estimates amounts to only 0.1 mm a$^{-1}$ (Table 2). As discussed, any GIA signal over the LPZ wod be removed erroneously in the method of Gunter et al. (2014), but the uncertainty in low-degree harmonics is assumed to be much higher than a potential GIA signal within the LPZ. The bias correction regionalises the





GIA estimate, i.e. derived mass changes always refer to the mean LPZ mass change. Table S1 illustrates the large effect on ice-mass change estimates depending on the four options of applying bias corrections: (1) debiased GIA-signal and debiased total-mass signal (Fig. 6, Eq. 15), (2) debiased GIA-signal and biased total-mass signal, (3) biased GIA-signal and debiased total-mass signal, and (4) biased GIA-signal and biased total-mass signal. In addition, Fig. S7 illustrates the results from option (4). The bias correction defines how the total-mass change is decomposed into mass signals and it is a strong constraint to

determine meaningful mass estimates out of the combination approach. The large uncertainty introduced by degree-1 and $C_{20}$ is suppressed at the cost of global consistency.

The definition of the LPZ, as an area in which a very small apparent GIA-mass signal and ice-mass signal is expected, has several disadvantages: (1) The precipitation of the last 40 years is not directly linked to GIA. (2) Areas are included which show quite relevant GIA-induced BEC in forward models, e.g. close to the Ross Ice Shelf (Martín-Español et al., 2016b). (3)

The threshold for low precipitation is arbitrary and cannot be based on physical reasons in relation to GIA. Depending on the precipitation product used, a different area where the bias is estimated might be considered. (4) The LPZ is a large area in which even a low GIA effect can cause several Gt a$^{-1}$ apparent mass changes. (5) The LPZ bias correction does not allow for a simple transfer of the approach to Greenland or to a global framework. Nevertheless, the estimation over LPZ is at least one possibility to consider the presumably existing biases.

Figure 3 in Shepherd et al. (2012) show large differences in EAIS mass change estimates derived from satellite gravimetry and altimetry. In principle, the question of quantifying GIA in EAIS arises. For this discussion, the reader is referred to e.g. Whitehouse (2018), Whitehouse et al. (2019).

### 5.3   Sensitivity to altimetry product

The choice of the altimetry products has a major effect on the GIA estimate. Using ICESat-only and Envisat-only products leads

to a RMS$_{RE}$ of 1.1 and 0.8 mm a$^{-1}$, respectively (Table 2). Both missions use different observation methods and have different spatial coverage. The radar altimetry time series of Envisat is sampled monthly but only to a latitude of 81.5° South. ICESat uses laser altimetry and its polar gap is smaller (South of 86°). This regards the spatial sampling of Kamb Ice Stream where a dominant ice-dynamic signal is expected. ICESat's campaign-style temporal sampling (Sect. 3.1, Gunter et al. (2009)) may affect the trend estimation significantly. The MM-Altimetry product uses mainly observations from ICESat and Envisat for the

time period 2003-03/2009-10. The trend derived from the combination product (MM-Altimetry) shows a spatial discontinuity at the 81.5° latitude limit of Envisat coverage (Fig. S1A, Fig. 5A). We attribute this to the sparse time sampling of the ICESat mission. Our results show that the difference through various altimetry products does not vanish by applying the bias correction (Sect. 4.2, Table 2). Furthermore, differences in the spatial GIA pattern are remarkable in MBL and VL (Fig. S5F, G). The co-estimation of an annual-seasonal signal in altimetry only leads to small changes in the overall result (Sect. 4.2, RMS$_{RE}$:

0.1 mm a$^{-1}$) but is more consistent with processing of other data and models.



## 5.4 Firn-process assumptions and uncertainties

A crucial point in the combination approach is the case distinction for $\rho_\alpha$ (Eq. 9). As mentioned in Sect. 2.1, it only considers uncertainty of altimetry and the FDM and not for GRACE nor the SMB trends. The resulting map for $\rho_\alpha$ (Fig. 5A, Fig. S4) does not agree with predefined, physically sensible density maps and results in ice density where it is not reasonable to assume ice dynamics, e.g. in large areas of EAIS. It largely depends on used data sets (Fig. S4B, C). An alternative to the $\rho_\alpha$ approach

could be the formal approach shown in Eq. (8). Technically this would be correct. However, it results in a ice-density weight for the whole AIS. We are aware that this is not correct either because presumable processes in the firn layer are not completely considered by input data and models. We suggest the use of a predefined density mask similar to Riva et al. (2009), but with a predefined significance criterion for all input data sets. This would need further investigation.

We investigated the application of the $\rho_\alpha$ approach (Eq. 10) to assign height changes to ice dynamics or firn processes.

If a negative SEC is firn-related, but erroneously attributed to the density of ice by Eq. (10), this will lead to a higher ice-mass decrease assigned to altimetry. GRACE would sense the true smaller ice-mass decrease. Through combination of both this discrepancy in ice-mass change would be assigned to a positive GIA signal. We suppose this is qualitatively visible for ice-density-weighted regions in the EAIS (Fig. 5A, B), e.g. the sector between a longitude of 30° and 100°. Furthermore we suppose this erroneously introduced positive GIA Signal explains a part of the GIA bias.

The propagation of the empirically determined error patterns (EOF 1–3) of the firn-process models (Sect. 4.1) show small effects on the spatial pattern of inverse GIA estimates (Fig. S5I–K). The $\mathrm{RMS_{RE}}$ of EOF 1, EOF 2 and EOF 3 results is 0.5, 0.3 and 0.3 mm a$^{-1}$, respectively (Table 2). Note that this deviation results solely through differences in similar climate models using the same forcing data.

Uncertainties assumed in Gunter et al. (2014) for $\sigma_{j_{h_{\mathrm{firn}}}}$ are very small compared to our results (Sect. 4.1, Fig. 3). In addition,

any long-term trend in firn mass and firn thickness is ignored by the equilibrium assumption made by the firn modelling. SEC from Altimetry and the IMAU-FDM show major differences even with a different sign for some areas, e.g. AP, QML (Fig. 1A,C). These differences may indicate that the equilibrium assumption of the FDM (Sect. 3.3) is not fulfilled for those areas of the AIS, i.e. that firn-thickness changes occur over the whole modelling period.

## 5.5 Sensitivity to time interval

We also investigate a GIA solution derived from data sets over almost the entire GRACE period (2002-04/2016-08) and the approximately six-year period of CryoSat-2 overlapping with GRACE (2010-07/2016-08). The dependence of these estimates cannot be attributed to a single processing choice: On the one hand, different data sets are used (depending on assembled altimetry missions). On the other hand, cSMBA trends and FDM-derived SEC differ largely depending on the selected time interval (Sect. 3.3, Fig. S3). Ice-mass change estimates are very high for the time interval 2010-07/2016-08 if no bias corrections

or both bias corrections are applied (Table 2). Estimating the mass balance from debiased GIA-mass change and biased total-mass change for time periods 2002-04/2016-08 and 2010-07/2016-08 results in mass changes of AIS: -48 and -70 (total), 18 and 67 (GIA), and -66 and -137 (ice) Gt a$^{-1}$, respectively (not in Table 2). This ice-mass-change estimate for the time period



2010-07/2016-08 is of a similar order of magnitude as estimates published by Sasgen et al. (2019) spanning a range from 128.4 to 182.4 Gt a$^{-1}$ ice-mass loss. Note that Sasgen et al. (2019) use slightly different time periods: 2011-01/2017-06 and 2011-01/2016-06. Nevertheless, it indicates that, for this example, it is appropriated to only correct for GIA bias to receive comparable results.

We show that GIA estimates depend on the used time period. From this we conclude further investigation is needed for an improved consideration of inter-annual firn variations.

### 5.6 Combination on time-series level

The combination of time series leads to similar results compared to the trend-based approach (referring to estimate from 2010-07 to 2016-08, Sect. 4.3). We combined time series only for this time period, where CryoSat-2 and GRACE data are available with monthly sampling and sufficient spatial coverage. A closer examination of time series is the aim of ongoing research. There is a need to account for monthly uncertainties in all input data sets which result e.g. from modelling assumptions. As it is the case for the combination on trend level, challenges are: (1) Consideration of uncertainties of all data sets, (2) differences in spatio-temporal sampling of both sensors, and (3) merging resolution discrepancies including the consideration of signal leakage in GRACE observations. For further discussion of challenges combining geodetic data on time-series level the reader is referred to e.g. King et al. (2006). In addition to the simple summation of time series, state space approaches in geodetic Earth system research show promising results, e.g. time-varying trends in GRACE and GNSS (Didova et al., 2016) as well as tide gauges (Frederikse et al., 2016). This may receive more attention once the first results of GRACE-FO and ICESat-2 missions are available soon.

### 6 Conclusions

We investigated a combination method to isolate the GIA signal from satellite gravimetry and altimetry data. We based this work on Gunter et al. (2014) as an example for inverse estimation of GIA-induced BEC. We investigated the sensitivity of this approach (Eq. 9) to the variation of input parameters (Table 1): (1) Degree-1 and $C_{20}$-products in satellite gravimetry, (2) different satellite altimetry products, (3) empirically determined errors of firn-process models (SMB and FDM), and (4) the use of different time epochs including diverse data. (5) Furthermore, the sensitivity to the combination on time-series level (Eq. 18) was investigated. For this purpose, time series rather than trends of the input data were combined.

The comparison between the data sets used in this study show impressive similarities in terms of the spatial pattern of determined trends (Fig. 1), given that the results of altimetry, gravimetry and the FDM are independent. The separation of GIA and ice-mass signals following Gunter et al. (2014) depends strongly on the input parameters and processing steps (Table 2).

As done by Gunter et al. (2014) ADC from gravimetry are treated differently for (1) estimating the GIA signal and (2) determining the mass balance (Sect. 2.2). (1) A Gaussian filter and destriping filter is applied to ADC from gravimetry. This predetermines the smoothness of the GIA solution. The GIA-induced BEC is calibrated over the LPZ (LPZ-based GIA bias correction) and converted to mass change by an effective density mask. (2) GRACE derived ADC is calibrated over the LPZ,



too (LPZ-based GRACE bias correction). The mass balance is the difference between the debiased total-mass change and the debiased GIA-mass change. The estimated biases and the Gaussian filtering is an implementation of *a priori* information to regionally constrain the GIA solution and the mass balance to Antarctica. We conclude that the LPZ-based bias correction is a very serious leverage to receive reasonable mass-change estimates (Fig. 6, S7, Table 2, S1).

The modification of the formal approach of the combination strategy (Eq. 8) using the estimation of $\rho_\alpha$ (Eq. 10) does not

lead to a physically evident pattern to account for processes in the firn and ice layer (Fig. 5A, S4). Furthermore it is sensitive to input data sets. We suggest to use predefined density maps with significance criterion accounting for all input data sets.

A crucial point of the combination approach are the limits of both geodetic satellite sensors. On the one hand, altimetry enables the derivation of SEC with a high resolution. However, observations are missing in some areas, e.g. valleys, coastal regions. Especially ice dynamics will take place in those areas and therefore are partly missing in altimetry-derived SEC. On

the other hand, GRACE records all mass changes, however at lower resolution and with a lower signal-to-noise ratio. Since the availability of the MM-Altimetry from Schröder et al. (2019), 14 years of used GRACE observations are now the time-limiting factor. This is expected to be extended with GRACE-FO (and bridging solutions).

Our sensitivity-analysis results of the integrals over the AIS with a buffer zone of 400 km are: (1) The use of different degree-1 and $C_{20}$ products in GRACE processing leads to biased total-mass changes from -43 to 25 Gt a$^{-1}$. The LPZ-based bias

corrections almost completely eliminates the effect on the GIA estimate (RMS$_{RE} \leq 0.1$ mm a$^{-1}$) and on derived mass-change estimates. (2) Results using different altimetry products show a spread for apparent GIA-mass change of 15 Gt a$^{-1}$ if applying the GIA bias correction. The spread is 30 Gt a$^{-1}$ without applying a bias correction. (3) The uncertainty patterns empirically estimated from the firn-process models generate a spread of debiased and biased GIA-mass estimates of 7 and 21 Gt a$^{-1}$, respectively. (4) The spread of GIA-mass change estimated over other time intervals is 49 (debiased) and 81 Gt a$^{-1}$ (biased).

(5) The debiased GIA-mass change derived by the combination on time-series level is 28 Gt a$^{-1}$ smaller than the corresponding trend-based estimate.

Our results do not fully address the uncertainty introduced by input parameters, e.g. through the assumed equilibrium state of the used firn model. In future work improvement is needed for the correction of apparent biases and for separation of processes in the firn and the ice layer. This will allow to combine the satellite observations to estimate a globally consistent inverse GIA

solutions on time-series level.

*Author contributions.* M. O. Willen and M. Horwath conceptualised the study. M. O. Willen performed the investigation, computation, and visualisation tasks, and wrote the manuscript. L. Schröder provided the ice altimetry time series. A. Groh supported with GRACE data processing and data combination advice. S. R. M. Ligtenberg, P. Kuipers Munneke and M. R. van den Broeke provided the SMB output from RACMO2.3p2, the FDM, and assistance in their uncertainty characterisation. All co-authors discussed and improved the manuscript.



*Competing interests.*  Michiel R. van den Broeke is a member of the editorial board of the journal. The authors declare that they have no
conflict of interest.

*Acknowledgements.*  We thank Cécile Agosta (Université de Liège, Belgium) for providing the SMB output and density fields of the MAR
model. We acknowledge Olga Engels (Universität Bonn, Germany) for discussion on details of data combination strategies. This work was
5   from the Deutsche Forschungsgemeinschaft (DFG) as part of the Special Priority Program (SPP)-1889 "Regional Sea Level Change and
Society" (SeaLevel). We would like to thank the German Space Operations Center (GSOC) of the German Aerospace Center (DLR) for
providing continuously and nearly 100 % of the raw telemetry data of the twin GRACE satellites.



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
