# Peer review of "Sensitivity of inverse glacial isostatic adjustment estimates over Antarctica"

_The Cryosphere, 2019_

## Referee Comment (RC1) · Anonymous Referee #1 · 18 Jun 2019

Review of "Sensitivity of inverse glacial isostatic adjustment estimates over Antarctica" by Matthias O. Willen et al., submitted to The Cryosphere Discussions.

Summary:

The authors present inverse GIA estimates over Antarctica derived from a combination of satellite altimetry and gravimetry. Specifically, the authors investigate the sensitivity of such inverse GIA estimates to a suite of different parameters such as choice of altimeter record, surface mass balance model / firn-proces models, and degree-1 and C20 products. An important component of their method is the bias correction applied (the mean GIA and GRACE trend is set to zero over the low precipitation zone in East Antarctica). The authors present the spread of GIA estimates caused by the different variables and conclude that the choice of altimetry product poses the largest

uncertainty on the debiased mass-change estimates.

General comments:

The manuscript presents a rigorous sensitivity assessment, and the results clearly show that it is problematic to simply assume one specific model/data set/ product in this kind of analysis. Clearly the spread between different (equally valid) models/products can be larger than the uncertainty claimed for each product. This is a very important conclusion. The manuscript is well written, and the figures are clear and illustrative. The complex study setup with many variables makes the manuscript a bit challenging to read though. Therefore, one recommendation for the authors is to consider if there is a way to present the results from Table 2 in a figure instead. In summary, I find that the methods applied are sound and robust, the manuscript well written and the results important, making the work worth publishing I have listed below some specific recommendation that I think the authors should address before the manuscript is published.

Specific comments:

p. 1, l. 1: strictly speaking there is also a bottom-melt term in the mass balance equation, even though it might be very small.

When you discuss firn it seems to me that you think of firn processes = SMB (e.g. p. 1, l. 35). Do you not differ between firn and snow? I would thinks that part of the SMB signal (on short temporal resolution) is caused by changes in snow and not firn and therefore your definitions confuse me. Please clarify this.

You argue that you can characterize the uncertainty of the SMB by comparing two models (RACMO and MAR), but do your results not imply that this might not be sufficient? The variability between those two are so large (fig 2) that it would seem very relevant to include more models. Please comment on this. Maybe no other models are available?

Are the errors you mention in line 4, page 3 actually errors?

In eq. 10, please explain the case of $\alpha = 0$. I do not understand the physical meaning
of this. Why is assuming 0 a better choice?

Your results are dependent on some assumptions, one of which is that the only region in Antarctica that experiences glacial thickening is the Kamb Ice Stream. I think that this is an important assumption. Can you back it up by more references?

I find it a bit strange to state that one of your aims is to reproduce the method of Gunter et al., (2014). It might be something you have to do for you to reach another aim, but I don't see it as your aim to reproduce previous results (p. 5, l. 16).

Regarding you assumptions on GIA-induced BEC: Please specify what threshold you use to define what is negligible. Also is there some references to back up you assumption that it is indeed negligible in the LPZ. (p. 5, l. 21-23).

Can you please elaborate on why a consistent filtering of the quotient is not possible? (p.6 ,l.8). Is an ocean leakage mass signal of 4.5Gt/year not relevant to take into account? (p. 6,l. 13) .

The sentence in p. 7., line 13-14 seems dis-tached from the rest. Can you elaborate a little on what the implications of such low viscosity areas are for your study?

Can you please explain why the altimetry combined time series differs in spatial coverage? I understand why they may be different from one mission to another but why from month to month? Due to data loss in some areas?

Please back up the statements that the ITSG-Grace2016 has the highest s-t-n ratio with some reference(s).

There is provided no explanation for why you choose a different annual precipitation as threshold for low precip. than what was used in Gunter et al., 2014.

Technical issues

p.2, l.3 : on Earth -> on and in Earth

p.2 , l. 7: -> .. through glaciations and deglaciations during the last..

p.2, l. 31 : are beyond -> are larger that the

p.3., l. it is explained -> we explain

Fig.2 : Clarify which altimetry product is visualized
* * *

---

## Referee Comment (RC2) · Anonymous Referee #2 · 12 Jul 2019

The manuscript by Willen et al. entitled "Sensitivity of inverse glacial isostatic adjustment estimates over Antarctica" explores the impact that various assumptions and input data sets have on the estimation of Antarctic GIA and ice mass change. The authors follow the methodology outlined in Gunter et al., 2014, but expand the analysis to include additional and updated data sets, such as the inclusion of additional altimetry measurements from Envisat and Cryosat-2, an updated RACMO climate model (v2.3), and an extended GRACE time series. The manuscript is well written, and is clear in describing the processing steps. Overall, it's a useful study and moves the state-of-the-art on this topic. That said, there were a handful of concerns/comments I came across while going through the manuscript. Below is a summary of the major and minor items, which I would kindly ask the authors to respond to:

Major items: ————

1) The GRACE time series, regardless of processing center, are relatively consistent, so there is little variation in this input data set. The SMB data sets show some significant variation (Fig. 2), but the limitation here is that firn height estimates can only be computed from the RACMO model, and not from the MAR model. If I understood things correctly, the authors do perform an EOF analysis on the model differences, and then use these to generate uncertainty estimates for RACMO. It was unclear exactly how this was done, so I think it would help to expand on this in the text (end of p. 12). How exactly are the errors added (sqrt sum of squares of each EOF sigma at each grid cell point)? And was the same approach applied to the hdot_firn term? If so, is this realistic, since firn compaction works over longer time scales and may be non-linear? I also didn't completely follow the statement at the top of p. 15 regarding the creation of 32 separate GIA estimate from 32 different trend estimates. Did you, for example, take a trend difference from one of the 32, 7-yr windows, add that to the nominal RACMO trend, and then calculate a GIA solution?

2) An alternative, and perhaps more complete, assessment of the influence of the SMB models might be to run the combination analysis without the altimetry inputs. The altimetry only serves to update potential mismodeling in the SMB estimates, and to identify areas of glacial thinning. A fixed map of regions of glacial thinning could be developed, e.g., derived from published surface velocity plots, and used to remove the regions with ice density. This thinning map would only need to be representative, since the purpose is only to examine the sensitivity of the SMB inputs. Then, if you use the same GRACE time series, this would essentially isolate the contribution of the SMB model on the combination. And it would show what a combination with RACMO and MAR would look like in a side-to-side comparison.

3) The treatment of the altimetry data was a concern for me. The reference altimetry product was the multi-mission (MM), but no plots are shown of the default uncertainty estimates of the trends for this data set, although the authors do mention these uncertainties are used in the combination. Furthermore, the altimetry does not appear to be calibrated to the LPZ like the other data sets. Without this, any reference frame offsets or other biases from the altimetry data will find their way into the combination solution. This is why the GIA and GRACE LPZ is implemented, and is why Gunter et al 2014 also estimated their ICESat biases over the same LPZ.

4) Following on the prior point, the application of the density term, rho_alpha, in the combination is going to be directly affected by the altimetry product (as recognized by the authors). An inspection of the density map in Fig 5 shows very few areas that appear to have values of zero. This suggests that for nearly all of Antarctica, including most of EA and the LPZ, the difference between the altimetry and FDM heights was > 2-sigma. This means that very few regions used the default mdot_firn value from the SMB model. Referencing Gunter et al, 2014, they note that the classification of the rho_alpha term was used to "only deal with potential residual signal observed between ICESat and the FDM. The majority of the surface mass changes come directly from the SMB estimates (i.e., mdot_firn) derived from RACMO2." See also their Fig 7, which shows where the dominant positive differences are found, which are limited to a few near-coastal regions. Is this also the case for the current study? There was not a difference map between the MM and FDM trends, so it's unclear whether the 2-sigma difference was large or small (and does this difference show near-zero change over the LPZ?).

5) It also appears that the MM altimetry is heavily influenced by the Envisat processing, as the density maps in the supplement (Fig S4) for the Envisat and MM look nearly identical. The ICESat density maps shows much more zero-density values. The Envisat altimetry shows large areas of EA (see e.g., the Dome Fuji region) with negative surface height change compared to the FDM, so these large areas are assigned a density of ice (917 kg/m3). Based on Fig 1., GRACE does not see mass loss in that region, so in the combination this difference is estimated to be GIA. This is why there is a large positive uplift seen in the Dome Fuji region. This positive BEC feature may

just be due a processing artifact of the Envisat data (e.g., an atmospheric correction or penetration bias, as described by Remy et al, 2014). It is these types of differences that I believe led the authors to state in their conclusions that using the rho_alpha criteria "does not lead to a physically evident pattern to account for processes in the firn and ice layer (Fig. 5A, S4). Furthermore it is sensitive to input data sets. We suggest to use predefined density maps with significance criterion accounting for all input data sets" (p. 22, ln 5). This raises some interesting points. First, the combination will always be sensitive to the input data sets – that's the nature of real-data combinations. It may be that the patterns seen are products of the input data sets, and not the combination methodology, and the solution will only improve when those input data sets are refined (to include GRACE, altimetry, climate data, etc.). Second, if a predefined map is used to designate regions of ice loss or unmodeled accumulation, then you might be forcing the data into a predefined result. And, what other data input would be used to generate this new map? It wasn't clear to me how this alternative approach would work, and what improvement it might have. Perhaps the authors can provide a sample case in which the suggested predefined density mask is used, and how this compares with the reference case. It's worth noting that Gunter et al 2014 do use a predefined density map similar to the Riva 2009 when assigning densities to the positive ICESat-FDM height changes > 2-sigma. It is only if this height change is negative and > 2-sigma that the density of ice is used, since it is assumed that such large negative height changes are due to ice loss.

6) A Cryosat-2 elevation trend map is not provided, but is a critical component to Sec 5.5 and 5.6, which claim that the combination approach is sensitive to the time interval used. No maps of the corresponding density for Cryosat-2 are provided either. Some mass change values are provided, and a match to Sasgen et al 2019 is implied, but only when the GRACE LPZ bias is ignored, but presumably with the GIA LPZ biased used. All of this does not provide very strong support for the claim that "GIA estimates depend on the used time period" (p. 21, ln 4). I would argue that as long as the input data is accurate, the time period shouldn't matter.
7) At several points in the paper, the authors present findings from a mixture of biased and dedebiased data sets. One example is in Sec 5.5 (p. 21, ln 4). Table S1 is another example. I can see the value in showing the magnitudes of the bias estimates, but a mass change result from, e.g., a debiased GIA solution and a biased GRACE solution, seems inconsistent. I would think you should only present either a fully biased or debiased solution to stay consistent. Otherwise, the various frame, deg1, and C20 biases get mixed differently depending on the combination chosen, and the solution becomes a mixture of global and regionally-constrained data.

8) Modeling the elastic correction as a constant scale factor (pg 9, ln 15) of the altimetry height change may introduce error, especially in regions such as the AP and ASE (where thinning and accumulation are significant). Were the actual magnitudes of these elastic corrections investigated? And how would these BEC corrections be distinguished from large viscoelastic responses suspected in these same regions?

9) While the authors are clear that it is a sensitivity study, there is no validation of the results (Gunter et al 2014 used GPS site displacements), so there is no assessment as to whether the variations observed by changing the input data sets are an improvement or not.

Minor items: ————

10) p. 12, ln 5: Just to clarify, the uncertainty you refere to here is the uncertainty as derived by the IMAU group for the mdot_firn term?

11) p. 12, ln 9: The 7-year window seems arbitrary. Why not 10 or 5 or 3 yrs? And, how does EOF analysis vary if another window timeframe is chosen?

12) p. 21, ln 4: certainly a longer period will result in more reliable results, but if the inputs are correct, the time period shouldn't matter. This statement should be rephrased to say that it depends on the quality of the input data sets.

---

## Editor Comment (EC1) · Pippa Whitehouse (Editor) · 16 Jul 2019

I would like to thank both reviewers for their detailed and constructive comments on this manuscript. A number of technical issues are highlighted in their comments, but both reviews are generally positive and therefore I encourage the submission of a revised manuscript.

Kind regards,

Pippa Whitehouse
* * *

---

## Author Comment (AC1) · 27 Sep 2019

The referee comments are enclosed with accents and indicated in italics. Blue text is used to indicate the author's response and changes in the manuscript.

**General comments**

*"The manuscript presents a rigorous sensitivity assessment, and the results clearly show that it is problematic to simply assume one specific model/data set/product in this kind of analysis. Clearly the spread between different (equally valid) models/products can be larger than the uncertainty claimed for each product. This is a very important conclusion. The manuscript is well written, and the figures are clear and illustrative.*

*The complex study setup with many variables makes the manuscript a bit challenging to read though. Therefore, one recommendation for the authors is to consider if there is a way to present the results from Table 2 in a figure instead. In summary, I find that the methods applied are sound and robust, the manuscript well written and the results important, making the work worth publishing I have listed below some specific recommendation that I think the authors should address before the manuscript is published."*

Thank you for your positive and constructive feedback. In Table 1 we provide an overview of all experiments and the used input for the sensitivity analysis. Fig. 6 and Fig. S7 illustrate the debiased and biased results from Table 2.

**Specific comments**

*"p. 1, l. 1: strictly speaking there is also a bottom-melt term in the mass balance equation, even though it might be very small."*

We added basal melt in the text.

*"When you discuss firn it seems to me that you think of firn processes = SMB (e.g. p.1, l. 35). Do you not differ between firn and snow? I would thinks that part of the SMB signal (on short temporal resolution) is caused by changes in snow and not firn and therefore your definitions confuse me. Please clarify this."*

We do not take into account a separate snow layer such as Zammit-Mangion et al. (2015). We summarise mass and volume changes of the ice sheet which do not take place in the ice layer with the term *firn processes*. We clarified this in the introduction of the manuscript.

*"You argue that you can characterize the uncertainty of the SMB by comparing two models (RACMO and MAR), but do your results not imply that this might not be suffi-cient? The variability between those two are so large (fig 2) that it would seem very*

[Figure]

*relevant to include more models. Please comment on this. Maybe no other models are available?"*

Thank you for pointing on this as the uncertainty characterisation of the models is a challenging task. At the moment, there is no rigorous study on this topic. It would be very worthwhile to use an ensemble of climate model products for this, e.g. by forcing the climate models with changing input parameters. But they are highly computational expensive. RACMO2 and MAR are the only two regional climate models which are comparable with regard to forcing, time period, and spatial coverage. We added this limitation at the end of Sect. 2.4.

*"Are the errors you mention in line 4, page 3 actually errors?"*

Since the models do not provide uncertainties, we argue that the differences of the model outputs represent the uncertainty of them. We explain the characterisation in Sect. 2.4 and 4.1.

*"In eq. 10, please explain the case of $\alpha$ = 0. I do not understand the physical meaning of this. Why is assuming 0 a better choice?"*

The case of $\alpha$ = 0 is used if the $2\sigma_h$-criterion is not reached. In this case $\dot{m}_{GIA} = \dot{m}_{grav} - \dot{m}_{firn}$. This means that no mass change in the ice layer is considered and mass changes of the ice sheet are fully described by the modelled trend of cumulated surface mass balance anomalies. We extended the explanation of this case in Sect. 2.1.

*"Your results are dependent on some assumptions, one of which is that the only region in Antarctica that experiences glacial thickening is the Kamb Ice Stream. I think that this is an important assumption. Can you back it up by more references?"*

We added the references Retzlaff and Bentley (1993) and Wingham et al. (2006).

*"I find it a bit strange to state that one of your aims is to reproduce the method of Gunter et al., (2014). It might be something you have to do for you to reach another aim, but I don't see it as your aim to reproduce previous results (p. 5, l. 16)."*

We agree and removed the corresponding sentence.

*"Regarding you assumptions on GIA-induced BEC: Please specify what threshold you use to define what is negligible. Also is there some references to back up you assumption that it is indeed negligible in the LPZ. (p. 5, l. 21-23)."*

A GIA-induced BEC in this area is predicted by GIA models from approximately -3 until +1 mm/a (Whitehouse et al., 2019). We do not define a threshold. Gunter et al. (2014) argue, "if any genuine GIA over the LPZ does exist, then this would erroneously bias the empirically derived rates from the combination approach; however, as mentioned already, any error of this kind is believed to be much lower than that introduced by the various other (imprecisely known) bias contributors." We discussed in Sect. 5.2 that the LPZ-based bias correction is a limitation of the combination approach. Further we extended Sect. 2.2 that the assumption of neglecting a small GIA-induced BEC in the LPZ introduces error.

*"Can you please elaborate on why a consistent filtering of the quotient is not possible? (p.6 ,l.8). Is an ocean leakage mass signal of 4.5Gt/year not relevant to take into account? (p. 6,l. 13)."*

A consistent filtering is not possible because we do not have access to an unfiltered $\dot{m}_{grav}$. GRACE derived monthly gravity field solutions are available with a theoretical spatial resolution of 150–300 km (Wouters et al., 2014), which is much less than Altimetry and the firn-process models with a resolution of roughly 10 km and 30 km, respectively. Furthermore, a filtering (smoothing) of the gravity field is unavoidable, because of the dominant error pattern. In the quotient this would be weighted with the high-resolution density mask. As we do not evaluate the sensitivity of filtering, the

ocean-leakage mass signal is the same in every experiment. We clarified this in the manuscript.

*"The sentence in p. 7., line 13-14 seems distached from the rest. Can you elaborate a little on what the implications of such low viscosity areas are for your study?"*

If there is a very low viscosity the assumption of a linear GIA-induced BEC introduce error. We extended the paragraph in the manuscript.

*"Can you please explain why the altimetry combined time series differs in spatial coverage? I understand why they may be different from one mission to another but why from month to month? Due to data loss in some areas?"*

The combined altimetry time series is compiled from observations of various altimetry missions. For example, ICESat and Envisat observed parallelly. Whereas ICESat has a higher spatial coverage than Envisat (polar gap), but only with a campaign-style temporal sampling. As a result the combined monthly-sampled time series has a higher spatial coverage during the months with observations from ICESat and Envisat. Further, as you mentioned, differing data quality through time is another reason. We extended the corresponding paragraph in Sect. 3.1.

*"Please back up the statements that the ITSG-Grace2016 has the highest s-t-n ratio with some reference(s)."*

We added the reference Jean et al. (2018). Therein ITSG-Grace2014 shows the lowest noise level. We rephrased the paragraph, because release 6 solutions including ITSG-Grace2018 presumably show a higher signal-to-noise ratio.

*"There is provided no explanation for why you choose a different annual precipitation as threshold for low precip. than what was used in Gunter et al., 2014."*

We used the 20 mm/a threshold used in Riva et al. (2009). We added the reference in the manuscript.

**Technical issues**

*"p.2, l.3 : on Earth → on and in Earth"*

*"p.2 , l. 7: → .. through glaciations and deglaciations during the last.."*

*"p.2, l. 31 : are beyond → are larger that the"*

*"p.3., l. it is explained → we explain"*

*"Fig.2 : Clarify which altimetry product is visualized"*

We implemented all suggestions.

**References**

Gunter, B., Didova, O., Riva, R. E. M., Ligtenberg, S. R. M., Lenaerts, J. T. M., King, M. A., van den Broeke, M. R., and Urban, T.: Empirical estimation of present-day Antarctic glacial isostatic adjustment and ice mass change, The Cryosphere, 8, 743–760, https://doi.org/10.5194/tc-8-743-2014, 2014.

Retzlaff, R. and Bentley, C.: Timing of stagnation of Ice Stream C, West Antarctica, from short-pulse radar studies of buried surface crevasses, J. Glac., 39, 553–561, 1993.

Riva, R. E. M., Gunter, B. C., Urban, T. J., Vermeersen, B. L. A., Lindenbergh, R. C., Helsen, M. M., Bamber, J. L., van de Wal, R. S. W., van den Broeke, M. R., and Schutz, B. E.: Glacial Isostatic Adjustment over Antarctica from combined ICESat and GRACE satellite data, Earth Planet. Sci. Lett., 288, 516–523, https://doi.org/10.1016/j.epsl.2009.10.013, 2009.

Whitehouse, P., Gomez, N., King, M., and Wiens, D.: Solid Earth change and the evolution of the Antarctic Ice Sheet, Nature Communications, 10, https://doi.org/10.1038/s41467-018-08068-y, 2019.

Wingham, D., Shepherd, A., Muir, A., and Marshall, G.: Mass balance of the Antarctic ice sheet, Phil. Trans. R. Soc. Lond. A, 364, 1627–1635, 2006.

Wouters, B., Bonin, J. A., Chambers, D. P., Riva, R. E. M., Sasgen, I., and Wahr, J. GRACE, time-varying gravity, Earth system dynamics and climate change. Reports on Progress in Physics, 77(11), 116801. https://doi.org/10.1088/0034-4885/77/11/116801, 2014.

Zammit-Mangion, A., Rougier, J., Schön, N., Lindgren, F., and Bamber, J.: Multivariate spatio-temporal modelling for assessing Antarctica's present-day contribution to sea-level rise, Environmetrics, 26, 159–177, https://doi.org/10.1002/env.2323, 2015.
* * *

---

## Author Comment (AC2) · 27 Sep 2019

The referee comments are enclosed with accents and indicated in italics. Blue text is used to indicate the author's response and changes in the manuscript.

We thank you for the positive comment and your suggestion to overcome the deficiencies of the manuscript.

**Major items**

*"1) The GRACE time series, regardless of processing center, are relatively consistent, so there is little variation in this input data set. The SMB data sets show some significant variation (Fig. 2), but the limitation here is that firn height estimates can only*

[Figure]

*be computed from the RACMO model, and not from the MAR model. If I understood things correctly, the authors do perform an EOF analysis on the model differences, and then use these to generate uncertainty estimates for RACMO. It was unclear exactly how this was done, so I think it would help to expand on this in the text (end of p. 12). How exactly are the errors added (sqrt sum of squares of each EOF sigma at each grid cell point)? And was the same approach applied to the hdot_firn term? If so, is this realistic, since firn compaction works over longer time scales and may be non-linear? I also didn't completely follow the statement at the top of p. 15 regarding the creation of 32 separate GIA estimate from 32 different trend estimates. Did you, for example, take a trend difference from one of the 32, 7-yr windows, add that to the nominal RACMO trend, and then calculate a GIA solution?"*

We used differences of estimated trends of cumulated surface mass balance anomalies (cSMBA). We assume those differences representing (a part of) the error of regional climate modelling. Unfortunately there is only the IMAU-FDM forced with RACMO2 outputs and no equivalent FDM forced with MAR outputs. For this reason we cannot directly get trend differences of firn thickness trends from two models. At the end of section 3.3 we explain how we estimate pseudo trend differences of firn thickness trends using density fields from MAR. We stated in the manuscript that this does not consider the correct evolution of the firn layer. The EOF analysis is done with the cSMBA trend differences. The normalised EOF is scaled with the square root of the particular eigenvalue (sigma). For the propagation towards the combination approach a pseudo firn thickness EOF is estimated using MAR density fields. The first three EOFs are added separately to the estimated cSMBA trend and firn thickness trend, respectively. We are aware of that this can only be a start of a rigorous uncertainty characterisation of climate model outputs and only consider a part of aspects. In the manuscript we extended Sect. 4.1 with a reference to Sect. 3.3 where we explain how pseudo EOFs and trend differences are computed. Yes, (1) we calculate the cSMBA trend difference, (2) calculate the pseudo firn thickness trend difference with the MAR density, and (3) add them to the nominal cSMBA and firn thickness trends, respectively.

With those updated trends we estimate a GIA solution. We do this for every trend difference resulting in 32 GIA solutions. We clarified this at the end of section 4.1.

*"2) An alternative, and perhaps more complete, assessment of the influence of the SMB models might be to run the combination analysis without the altimetry inputs. The altimetry only serves to update potential mismodeling in the SMB estimates, and to identify areas of glacial thinning. A fixed map of regions of glacial thinning could be developed, e.g., derived from published surface velocity plots, and used to remove the regions with ice density. This thinning map would only need to be representative, since the purpose is only to examine the sensitivity of the SMB inputs. Then, if you use the same GRACE time series, this would essentially isolate the contribution of the SMB model on the combination. And it would show what a combination with RACMO and MAR would look like in a side-to-side comparison."*

In our study we focussed on the combination approach published by Gunter et al. (2014). From a historic point of view, Wahr et al. (2000) suggested the combination of satellite gravimetry and altimetry to isolate the GIA signal. Gunter et al. (2014) used climate model products to overcome some limitations of the geodetic satellite data. We agree it would be worthwhile investigating differing data/model combination strategies to isolate GIA. On the one hand, the suggested strategy would give more insights using climate model products in combination with GRACE-derived gravity fields. On the other hand, this investigation will increase the complexity of the study. This is a point of criticism from the first referee.

*"3) The treatment of the altimetry data was a concern for me. The reference altimetry product was the multi-mission (MM), but no plots are shown of the default uncertainty estimates of the trends for this data set, although the authors do mention these uncertainties are used in the combination. Furthermore, the altimetry does not appear to be calibrated to the LPZ like the other data sets. Without this, any reference frame offsets*

*or other biases from the altimetry data will find their way into the combination solution. This is why the GIA and GRACE LPZ is implemented, and is why Gunter et al 2014 also estimated their ICESat biases over the same LPZ."*

We extended Fig. S1 in the supplementary material with uncertainty maps of every used altimetry product. GRACE-derived area density changes are not calibrated to the LPZ prior the actual combination (Eq. 9). GRACE-derived area density changes and the GIA solution from the combination are calibrated over the LPZ to determine the mass balance. In other words: The combined result derived from GRACE, altimetry and firn process models, namely the GIA-induced BEC, is calibrated over the LPZ. Existing biases sum up in the combination and are jointly removed. We explained this in more detail in Sect. 2.2 in the manuscript. But still the calibration of the used altimetry products is different to Gunter et al. (2014). Gunter et al. (2014) uses the LPZ to estimate the ICESat campaign biases. This results in a zero trend of SEC over the LPZ. The campaign biases from Schröder et al. (2019) are calibrated with kinematic GNSS measurement over Lake Vostok. The inter-mission biases during relevant period are calibrated via overlapping observations. More details can be find in Schröder et al. (2019).

*"4) Following on the prior point, the application of the density term, rho_alpha, in the combination is going to be directly affected by the altimetry product (as recognized by the authors). An inspection of the density map in Fig 5 shows very few areas that appear to have values of zero. This suggests that for nearly all of Antarctica, including most of EA and the LPZ, the difference between the altimetry and FDM heights was > 2-sigma. This means that very few regions used the default mdot_firn value from the SMB model. Referencing Gunter et al, 2014, they note that the classification of the rho_alpha term was used to "only deal with potential residual signal observed between ICESat and the FDM. The majority of the surface mass changes come directly from the SMB estimates (i.e., mdot_firn) derived from RACMO2." See also their Fig 7, which shows where the dominant positive differences are found, which are limited to a few*

*near-coastal regions. Is this also the case for the current study? There was not a difference map between the MM and FDM trends, so it's unclear whether the 2-sigma difference was large or small (and does this difference show near-zero change over the LPZ?)."*

Fig. 7b in Gunter et al. (2014) shows differences between surface elevation changes derived from ICESat and the FDM. But only differences are shown which are greater than 6 cm a$^{-1}$. It is unclear to us why this threshold was used. Unfortunately, it is not shown where the difference is > 2-sigma. Unlike us, Gunter et al. (2014) do not show their $\rho_\alpha$ map which is used in the combination and would provide information where the difference is > 2-sigma. Fig. 1 (in this comment) shows the differences between ICESat and FDM we used as input without clipping. Those differences are small in EA. But those small differences are weighted with ice density, because they are > 2-sigma. In comparison, Fig. 1 shows the map published in Gunter et al. (2014), with the 6 cm a$^{-1}$ threshold.

*"5) It also appears that the MM altimetry is heavily influenced by the Envisat process-ing, as the density maps in the supplement (Fig S4) for the Envisat and MM look nearly identical. The ICESat density maps shows much more zero-density values. The En-visat altimetry shows large areas of EA (see e.g., the Dome Fuji region) with negative surface height change compared to the FDM, so these large areas are assigned a density of ice (917 kg/m3). Based on Fig 1., GRACE does not see mass loss in that region, so in the combination this difference is estimated to be GIA. This is why there is a large positive uplift seen in the Dome Fuji region. This positive BEC feature may just be due a processing artifact of the Envisat data (e.g., an atmospheric correction or penetration bias, as described by Remy et al, 2014). It is these types of differences that I believe led the authors to state in their conclusions that using the rho_alpha criteria "does not lead to a physically evident pattern to account for processes in the firn and ice layer (Fig. 5A, S4). Furthermore it is sensitive to input data sets. We suggest to use predefined density maps with significance criterion accounting for all input data sets"*

*(p. 22, ln 5). This raises some interesting points. First, the combination will always be sensitive to the input data sets – that's the nature of real-data combinations. It may be that the patterns seen are products of the input data sets, and not the combination methodology, and the solution will only improve when those input data sets are refined (to include GRACE, altimetry, climate data, etc.). Second, if a predefined map is used to designate regions of ice loss or unmodeled accumulation, then you might be forcing the data into a predefined result. And, what other data input would be used to generate this new map? It wasn't clear to me how this alternative approach would work, and what improvement it might have. Perhaps the authors can provide a sample case in which the suggested predefined density mask is used, and how this compares with the reference case. It's worth noting that Gunter et al 2014 do use a predefined density map similar to the Riva 2009 when assigning densities to the positive ICESat-FDM height changes > 2-sigma. It is only if this height change is negative and > 2-sigma that the density of ice is used, since it is assumed that such large negative height changes are due to ice loss."*

Our results demonstrate the strong sensitivity towards differing altimetry products. ICESat and Envisat are different with regard to observing technique, spatial and temporal coverage, and temporal sampling. We agree that the limitations are due to the quality of the data which was not clear enough in the manuscript. We improved this. The case distinction of $\rho_\alpha$ is made to cope with apparent limitations of the firn-thickness trends and altimetry derived trends instead of using the formal approach (Eq. 8). A further investigation of different combination strategies would be very beneficial. Including the aim to find a better combination methodology would make the study more complicated. To avoid this, we removed speculative phrases on possible improvements in the manuscript and focussing on the sensitivity assessment of the investigated approach.

*"6) A Cryosat-2 elevation trend map is not provided, but is a critical component to Sec 5.5 and 5.6, which claim that the combination approach is sensitive to the time interval used. No maps of the corresponding density for Cryosat-2 are provided either. Some*

*mass change values are provided, and a match to Sasgen et al 2019 is implied, but only when the GRACE LPZ bias is ignored, but presumably with the GIA LPZ biased used. All of this does not provide very strong support for the claim that "GIA estimates depend on the used time period" (p. 21, ln 4). I would argue that as long as the input data is accurate, the time period shouldn't matter."*

We agree that as long as the input data is "correct" there is no time dependency. As you mentioned, the true limitations of the input data is the reason for differing results. We rephrased the corresponding paragraphs. The Multi-Mission-Altimetry product is dominated by CryoSat-2 observations during the time period 2010-07/2016-08. We clarified this in the manuscript. Fig. S1F shows the SEC. We decided not to use an additional CryoSat-2 only experiment. Fig. 2 (in this comment) compares Multi-Mission-derived and CryoSat-2-only-derived SEC. As you already mentioned, also Fig. 2 shows that the Envisat processing influences the result.

*"7) At several points in the paper, the authors present findings from a mixture of biased and dedebiased data sets. One example is in Sec 5.5 (p. 21, ln 4). Table S1 is another example. I can see the value in showing the magnitudes of the bias estimates, but a mass change result from, e.g., a debiased GIA solution and a biased GRACE solution, seems inconsistent. I would think you should only present either a fully biased or debiased solution to stay consistent. Otherwise, the various frame, deg1, and C20 biases get mixed differently depending on the combination chosen, and the solution becomes a mixture of global and regionally-constrained data."*

We fully agree and only present completely biased or debiased estimates. We removed mixed values from the text and Table S1 from the supplementary material.

*"8) Modeling the elastic correction as a constant scale factor (pg 9, ln 15) of the altimetry height change may introduce error, especially in regions such as the AP and ASE (where thinning and accumulation are significant). Were the actual magnitudes*

*of these elastic corrections investigated? And how would these BEC corrections be distinguished from large viscoelastic responses suspected in these same regions?"*

The constant scale factor does introduce error but this is negligible (Riva et al., 2009, Groh et al., 2012). We pointed that out in Sect. 3.1. The strong Gaussian smoothing further mitigates the influence of this error because large local amplitudes are damped. For illustration Fig. 3 (in this comment) compares vertical elastic deformation rates calculated from smoothed Multi-Mission-altimetry trends. This is done (1) by modelling in the spatial domain and (2) by the constant scale factor of 1.5 %. For (1) we used a predefined density mask to estimate mass change rates and rheological parameters from PREM. The differences between (1) and (2) vary between approximately -0.1 to 1.0 mm a$^{-1}$.

*"9) While the authors are clear that it is a sensitivity study, there is no validation of the results (Gunter et al 2014 used GPS site displacements), so there is no assessment as to whether the variations observed by changing the input data sets are an improvement or not."*

As you mentioned, our aim is to address the sensitivity of an existing methodology from which we conclude limitations. The investigated approach might be inappropriate to judge the quality of input data sets with GNSS observations. All input data sets are combined with existing bias. This bias is jointly removed using the LPZ-based bias correction. During this step unknown systematic errors in the input data can cancel out each other. GNSS observations might judge those input data sets as an improvement.

**Minor items**

*"10) p. 12, ln 5: Just to clarify, the uncertainty you refere to here is the uncertainty as derived by the IMAU group for the mdot_firn term?"*

This is uncertainty we receive (1) from our least square estimation when we estimate

trends of cSMBA and (2) taking 10 % of the estimated cSMBA trend over the ICESat-observation period. We clarified this in the manuscript.

*"11) p. 12, ln 9: The 7-year window seems arbitrary. Why not 10 or 5 or 3 yrs? And, how does EOF analysis vary if another window timeframe is chosen?"*

We used a 7-year window, because this corresponds to the ICESat's observation period. We stated this in Sect. 3.3. The trend differences would increase if the time interval is shorter and decrease with a longer time window. Fig. 4 (in this comment) shows the first three EOFs using a 5 year, 7 year, and 10 year time interval, respectively. The dominant patterns remain with respect to spatial pattern and amplitude. Whereby the amount of the explained total variance changes.

*"12) p. 21, ln 4: certainly a longer period will result in more reliable results, but if the inputs are correct, the time period shouldn't matter. This statement should be rephrased to say that it depends on the quality of the input data sets."*

We agree and rephrased this statement.

**References**

Groh, A., Ewert, H., Scheinert, M., Fritsche, M., Rülke, A., Richter, A., Rosenau, R., and Dietrich, R.: An Investigation of Glacial Isostatic Adjustment over the Amundsen Sea sector, West Antarctica, Global Planet. Change, 98–99, 45–53, https://doi.org/10.1016/j.gloplacha.2012.08.001, 2012.

Gunter, B., Didova, O., Riva, R. E. M., Ligtenberg, S. R. M., Lenaerts, J. T. M., King, M. A., van den Broeke, M. R., and Urban, T.: Empirical estimation of present-day Antarctic glacial isostatic adjustment and ice mass change, The Cryosphere, 8, 743–760, https://doi.org/10.5194/tc-8-743-2014, 2014.

Riva, R. E. M., Gunter, B. C., Urban, T. J., Vermeersen, B. L. A., Lindenbergh, R. C., Helsen, M. M., Bamber, J. L., van de Wal, R. S. W., van den Broeke, M. R., and Schutz, B. E.: Glacial Isostatic Adjustment over Antarctica from combined ICESat and GRACE satellite data, Earth Planet. Sci. Lett., 288, 516–523, https://doi.org/10.1016/j.epsl.2009.10.013, 2009.

Schröder, L., Horwath, M., Dietrich, R., Helm, V., van den Broeke, M. R., and Ligtenberg, S. R. M.: Four decades of Antarctic surface elevation changes from multi-mission satellite altimetry, The Cryosphere, 13, 427–449, https://doi.org/10.5194/tc-13-427-2019, 2019.

Wahr, J., Wingham, D., and Bentley, C.: A method of combining ICESat and GRACE satellite data to constrain Antarctic mass balance, J. Geophys. Res., 105, 16 279–16 294, https://doi.org/10.1029/2000JB900113, 2000.

**Fig. 1.** Left: The differences between ICESat and FDM from input data sets we used. Right: The original figure from Gunter et al. (2014) with a threshold of 6 cm/a and the masked Kamb Ice Stream.

[Figure]

-200  -100   0   100   200   -200  -100   0   100   200   -100   -50   0   50   100
SEC in $mm\,a^{-1}$          SEC in $mm\,a^{-1}$          SEC in $mm\,a^{-1}$

**Fig. 2.** The comparison of Multi-Mission-Altimetry derived trends (left), CryoSat-2-only derived trends (middle), and the difference of Multi-Misson-CryoSat-2 (right). Note the different value range.

[Figure]

**Fig. 3.** A: Trends from MM altimetry (Gaussian smoothing, half response: 400 km). Elastic-induced BEC using modelling (B) and estimated with -1.5 % from A (C). D: The difference map between B and C.

[Figure]

Fig. 4. Comparison of the EOF analysis using different time intervals. A–C, D–F, and G–I show the first three EOFs estimated over 5-year, 7-year, and 10-year time interval, respectively.

---

## Author Response (AR1)

This document contains:

- Author comment to referee comment 1
- Author comment to referee comment 2
- Author comment to editor decision
- Marked-up manuscript version

The referee comments are enclosed with accents and indicated in italics. Blue and red text is used to indicate the author's response and changes in the manuscript, respectively.

**Author comment to referee comment 1**

**General comments**

*"The manuscript presents a rigorous sensitivity assessment, and the results clearly show that it is problematic to simply assume one specific model/data set/product in this kind of analysis. Clearly the spread between different (equally valid) models/products can be larger than the uncertainty claimed for each product. This is a very important conclusion. The manuscript is well written, and the figures are clear and illustrative. The complex study setup with many variables makes the manuscript a bit challenging to read though. Therefore, one recommendation for the authors is to consider if there is a way to present the results from Table 2 in a figure instead. In summary, I find that the methods applied are sound and robust, the manuscript well written and the results important, making the work worth publishing I have listed below some specific recommendation that I think the authors should address before the manuscript is published."*

Thank you for your positive and constructive feedback. In Table 1 we provide an overview of all experiments and the used input for the sensitivity analysis. Fig. 6 and Fig. S7 illustrate the debiased and biased results from Table 2.

**Specific comments**

*"p. 1, l. 1: strictly speaking there is also a bottom-melt term in the mass balance equation, even though it might be very small."*

We added basal melt in Sect. 1.

*"When you discuss firn it seems to me that you think of firn processes = SMB (e.g. p.1, l. 35). Do you not differ between firn and snow? I would thinks that part of the SMB signal (on short temporal resolution) is caused by changes in snow and not firn and therefore your definitions confuse me. Please clarify this."*

We take not into account a separate snow layer such as Zammit-Mangion et al. (2015). We summarise mass and volume changes of the ice sheet which do not take place in the ice layer with the term *firn processes*.

We clarified this in Sect. 1 of the manuscript.

*"You argue that you can characterize the uncertainty of the SMB by comparing two models (RACMO and MAR), but do your results not imply that this might not be sufficient? The variability between those two are so large (fig 2) that it would seem very relevant to include more models. Please comment on this. Maybe no other models are available?"*

Thank you for pointing on this as the uncertainty characterisation of the models is a challenging task. At the moment, there is no rigorous study on this topic. It would be very worthwhile to use an ensemble of climate model products for this, e.g. by forcing the climate models with changing input parameters. But they are highly computational expensive. RACMO2 and MAR are the only two regional climate models which are comparable with regard to forcing, time period, and spatial coverage.

We added this limitation at the end of Sect. 2.4.

*"Are the errors you mention in line 4, page 3 actually errors?"*

Since the models do not provide uncertainties, we argue that the differences of the model outputs represent the uncertainty of them.

We explain the characterisation in Sect. 2.4 and 4.1.

*"In eq. 10, please explain the case of $\alpha$ = 0. I do not understand the physical meaning of this. Why is assuming 0 a better choice?"*

The case of $\alpha$ = 0 is used if the $2\sigma_h$-criterion is not reached. In this case $\dot{m}_{GIA} = \dot{m}_{grav} - \dot{m}_{firn}$. This means that no mass change in the ice layer is considered and mass changes of the ice sheet are fully described by the modelled trend of cumulated surface mass balance anomalies.

We extended the explanation of this case in Sect. 2.1.

*"Your results are dependent on some assumptions, one of which is that the only region in Antarctica that experiences glacial thickening is the Kamb Ice Stream. I think that this is an important assumption. Can you back it up by more references?"*

We added the references Retzlaff and Bentley (1993) and Wingham et al. (2006) in Sect. 2.1 and 5.3.

*"I find it a bit strange to state that one of your aims is to reproduce the method of Gunter et al., (2014). It might be something you have to do for you to reach another aim, but I don't see it as your aim to reproduce previous results (p. 5, l. 16)."*

We agree and removed the corresponding sentence in Sect 2.2.

*"Regarding you assumptions on GIA-induced BEC: Please specify what threshold you use to define what is negligible. Also is there some references to back up you assumption that it is indeed negligible in the LPZ. (p. 5, l. 21-23)."*

A GIA-induced BEC in this area is predicted by GIA models from approximately -3 until +1 mm/a (Whihouse et al., 2019). We do not define a threshold. Gunter et al. (2014) argue, "if any genuine GIA over the LPZ does exist, then this would erroneously bias the empirically derived rates from the combination approach; however, as mentioned already, any error of this kind is believed to be much lower than that introduced by the various other (imprecisely known) bias contributors." We discussed in Sect. 5.2 that the LPZ-based bias correction is a limitation of the combination approach.

Further, we extended Sect. 2.2 that the assumption of neglecting a small GIA-induced BEC in the LPZ introduce error.

*"Can you please elaborate on why a consistent filtering of the quotient is not possible? (p.6 ,l.8). Is an ocean leakage mass signal of 4.5Gt/year not relevant to take into account? (p. 6,l. 13)."*

A consistent filtering is not possible because we do not have access to an unfiltered $\dot{m}_{grav}$. GRACE derived monthly gravity field solutions are available with a theoretical spatial resolution of 150–300 km (Wouters et al., 2014), which is much less than Altimetry and the firn-process models with a resolution of roughly 10 km and 30 km, respectively. Furthermore, a filtering (smoothing) of the gravity field is unavoidable, because of the dominant error pattern. In the quotient this would be weighted with the high-resolution density mask. As we do not evaluate the sensitivity of filtering, the ocean-leakage mass signal is the same in every experiment.

We clarified this in Sect. 3.

*"The sentence in p. 7., line 13-14 seems distached from the rest. Can you elaborate a little on what the implications of such low viscosity areas are for your study?"*

If there is a very low viscosity the assumption of a linear GIA-induced BEC introduce error.

We extended the paragraph in Sect. 2.5.

*"Can you please explain why the altimetry combined time series differs in spatial coverage? I understand why they may be different from one mission to another but why from month to month? Due to data loss in some areas?"*

The combined altimetry time series is compiled from observations of various altimetry missions. For example, ICESat and Envisat observed parallelly. Whereas ICESat has a higher spatial coverage than Envisat (polar gap), but only with a campaign-style temporal sampling. As a result the combined monthly-sampled time series has a higher spatial coverage during the months with observations from ICESat and Envisat. Further, as you mentioned, differing data quality through time is another reason.

We extended the corresponding paragraph in Sect. 3.1.

*"Please back up the statements that the ITSG-Grace2016 has the highest s-t-n ratio with some reference(s)."*

We added the reference Jean et al. (2018). Therein ITSG-Grace2014 shows the lowest noise level. We rephrased the paragraph in Sect. 3.2, because release 6 solutions including ITSG-Grace2018 presumably show a higher signal-to-noise ratio.

*"There is provided no explanation for why you choose a different annual precipitation as threshold for low precip. than what was used in Gunter et al., 2014."*

We used the 20 mm/a threshold used in Riva et al. (2009).

We added the reference in Sect 3.3.

**Technical issues**

*"p.2, l.3 : on Earth → on and in Earth"*

*"p.2 , l. 7: → .. through glaciations and deglaciations during the last.."*

*"p.2, l. 31 : are beyond → are larger that the"*

*"p.3., l. it is explained → we explain"*

*"Fig.2 : Clarify which altimetry product is visualized"*

We implemented all suggestions.

**Author comment to referee comment 2**

We thank you for the positive comment and your suggestion to overcome the deficiencies of the manuscript.

**Major items**

*"1) The GRACE time series, regardless of processing center, are relatively consistent, so there is little variation in this input data set. The SMB data sets show some significant variation (Fig. 2), but the limitation here is that firn height estimates can only be computed from the RACMO model, and not from the MAR model. If I understood things correctly, the authors do perform an EOF analysis on the model differences, and then use these to generate uncertainty estimates for RACMO. It was unclear exactly how this was done, so I think it would help to expand on this in the text (end of p. 12). How exactly are the errors added (sqrt sum of squares of each EOF sigma at each grid cell point)? And was the same approach applied to the hdot_firn term? If so, is this realistic, since firn compaction works over longer time scales and may be non-linear? I also didn't completely follow the statement at the top of p. 15 regarding the creation of 32 separate GIA estimate from 32 different trend estimates. Did you, for example, take a trend difference from one of the 32, 7-yr windows, add that to the nominal RACMO trend, and then calculate a GIA solution?"*

We used differences of estimated trends of cumulated surface mass balance anomalies (cSMBA). We assume those differences representing (a part of) the error of regional climate modelling. Unfortunately there is only the IMAU-FDM forced with RACMO2 outputs and no equivalent FDM forced with MAR outputs. For this reason we cannot directly get trend differences of firn thickness trends from two models. At the end of section 3.3 we explain how we estimate pseudo trend differences of firn thickness trends using density fields from MAR. We stated in the manuscript that this does not consider the correct evolution of the firn layer. The EOF analysis is done with the cSMBA trend differences. The normalised EOF is scaled with the square root of the particular eigenvalue (sigma). For the propagation towards the combination approach a pseudo firn thickness EOF is estimated using MAR density fields. The first three EOFs are added separately to the estimated cSMBA trend and firn thickness trend, respectively. We are aware of that this can only be a start of a rigorous uncertainty characterisation of climate model outputs and only consider a part of aspects. In the manuscript we extended Sect. 4.1 with a reference to Sect. 3.3 where we explain how pseudo EOFs and trend differences are computed. Yes, (1) we calculate the cSMBA trend difference, (2) calculate the pseudo firn thickness trend difference with the MAR density, and (3) add them to the nominal cSMBA and firn thickness trends, respectively. With those updated trends we estimate a GIA solution. We do this for every trend difference resulting in 32 GIA solutions. We clarified this at the end of Sect. 4.1.

*"2) An alternative, and perhaps more complete, assessment of the influence of the SMB models might be to run the combination analysis without the altimetry inputs. The altimetry only serves to update potential mismodeling in the SMB estimates, and to identify areas of glacial thinning. A fixed map of regions of glacial thinning could be developed, e.g., derived from published surface velocity plots, and used to remove the regions with ice density. This thinning map would only need to be representative, since the purpose is only to examine the sensitivity of the SMB inputs. Then, if you use the same GRACE time series, this would essentially isolate the contribution of the SMB model on the combination. And it would show what a combination with RACMO and MAR would look like in a side-to-side comparison."*

In our study we focussed on the combination approach published by Gunter et al. (2014). From a historic point of view, Wahr et al. (2000) suggested the combination of satellite gravimetry and altimetry to isolate the GIA signal. Gunter et al. (2014) used climate model products to overcome some limitations of the geodetic satellite data. We agree it would be worthwhile investigating differing data/model combination strategies to isolate GIA. On the one hand, the suggested strategy would give more insights using climate model products in combination with GRACE-derived gravity fields. On the other hand, this investigation will increase the complexity of the study. This is a point of criticism from the first referee.

*"3) The treatment of the altimetry data was a concern for me. The reference altimetry product was the multi-mission (MM), but no plots are shown of the default uncertainty estimates of the trends for this data set, although the authors do mention these uncertainties are used in the combination. Furthermore, the*

*altimetry does not appear to be calibrated to the LPZ like the other data sets. Without this, any reference frame offsets or other biases from the altimetry data will find their way into the combination solution. This is why the GIA and GRACE LPZ is implemented, and is why Gunter et al 2014 also estimated their ICESat biases over the same LPZ."*

GRACE-derived area density changes are not calibrated to the LPZ prior the actual combination (Eq. 9). GRACE-derived area density changes and the GIA solution from the combination are calibrated over the LPZ to determine the mass balance. In other words: The combined result derived from GRACE, altimetry and firn process models, namely the GIA-induced BEC, is calibrated over the LPZ. Existing biases sum up in the combination and are jointly removed. But still the calibration of the used altimetry products is different to Gunter et al. (2014). Gunter et al. (2014) uses the LPZ to estimate the ICESat campaign biases. This results in a zero trend of SEC over the LPZ. The campaign biases from Schröder et al. (2019) are calibrated with kinematic GNSS measurement over Lake Vostok. The inter-mission biases during relevant period are calibrated via overlapping observations. More details can be find in Schröder et al. (2019).

We extended Fig. S1 in the supplementary material with uncertainty maps of every used altimetry product. We explained the bias correction in more detail in Sect. 2.2.

*"4) Following on the prior point, the application of the density term, rho_alpha, in the combination is going to be directly affected by the altimetry product (as recognized by the authors). An inspection of the density map in Fig 5 shows very few areas that appear to have values of zero. This suggests that for nearly all of Antarctica, including most of EA and the LPZ, the difference between the altimetry and FDM heights was > 2-sigma. This means that very few regions used the default mdot_firn value from the SMB model. Referencing Gunter et al, 2014, they note that the classification of the rho_alpha term was used to "only deal with potential residual signal observed between ICESat and the FDM. The majority of the surface mass changes come directly from the SMB estimates (i.e., mdot_firn) derived from RACMO2." See also their Fig 7, which shows where the dominant positive differences are found, which are limited to a few near-coastal regions. Is this also the case for the current study? There was not a difference map between the MM and FDM trends, so it's unclear whether the 2-sigma difference was large or small (and does this difference show near-zero change over the LPZ?)."*

Fig. 7b in Gunter et al. (2014) shows differences between surface elevation changes derived from ICESat and the FDM. But only differences are shown which are greater than 6 cm a$^{-1}$. It is unclear to us why this threshold was used. Unfortunately, it is not shown where the difference is > 2-sigma. Unlike us, Gunter et al. (2014) do not show their $\rho_\alpha$ map which is used in the combination and would provide information where the difference is > 2-sigma. Fig. 1 (in this comment) shows the differences between ICESat and FDM we used as input without clipping. Those differences are small in EA. But those small differences are weighted with ice density, because they are > 2-sigma. In comparison, Fig. 1 shows the map published in Gunter et al. (2014), with the 6 cm a$^{-1}$ threshold.

*"5) It also appears that the MM altimetry is heavily influenced by the Envisat processing, as the density maps in the supplement (Fig S4) for the Envisat and MM look nearly identical. The ICESat density maps shows much more zero-density values. The Envisat altimetry shows large areas of EA (see e.g., the Dome Fuji region) with negative surface height change compared to the FDM, so these large areas are assigned a density of ice (917 kg/m3). Based on Fig 1., GRACE does not see mass loss in that region, so in the combination this difference is estimated to be GIA. This is why there is a large positive uplift seen in the Dome Fuji region. This positive BEC feature may just be due a processing artifact of the Envisat data (e.g., an atmospheric correction or penetration bias, as described by Remy et al, 2014). It is these types of differences that I believe led the authors to state in their conclusions that using the rho_alpha criteria "does not lead to a physically evident pattern to account for processes in the firn and ice layer (Fig. 5A, S4). Furthermore it is sensitive to input data sets. We suggest to use predefined density maps with significance criterion accounting for all input data sets" (p. 22, ln 5). This raises some interesting points. First, the combination will always be sensitive to the input data sets – that's the nature of real-data combinations. It may be that the patterns seen are products of the input data sets, and not the combination methodology, and the solution will only improve when those input data sets are refined (to include GRACE, altimetry, climate data, etc.). Second, if a predefined map is used to designate regions of ice loss or unmodeled*

*accumulation, then you might be forcing the data into a predefined result. And, what other data input would be used to generate this new map? It wasn't clear to me how this alternative approach would work, and what improvement it might have. Perhaps the authors can provide a sample case in which the suggested predefined density mask is used, and how this compares with the reference case. It's worth noting that Gunter et al 2014 do use a predefined density map similar to the Riva 2009 when assigning densities to the positive ICESat-FDM height changes > 2-sigma. It is only if this height change is negative and > 2-sigma that the density of ice is used, since it is assumed that such large negative height changes are due to ice loss."*

Our results demonstrate the strong sensitivity towards differing altimetry products. ICESat and Envisat are different with regard to observing technique, spatial and temporal coverage, and temporal sampling. We agree that the limitations are due to the quality of the data which was not clear enough in the manuscript. We improved this. The case distinction of $\rho_\alpha$ is made to cope with apparent limitations of the firn-thickness trends and altimetry derived trends instead of using the formal approach (Eq. 8). A further investigation of different combination strategies would be very beneficial. Including the aim to find a better combination methodology would make the study more complicated.
To avoid this, we removed the speculative sentence on possible improvements in Sect. 6.

*"6) A Cryosat-2 elevation trend map is not provided, but is a critical component to Sec 5.5 and 5.6, which claim that the combination approach is sensitive to the time interval used. No maps of the corresponding density for Cryosat-2 are provided either. Some mass change values are provided, and a match to Sasgen et al 2019 is implied, but only when the GRACE LPZ bias is ignored, but presumably with the GIA LPZ biased used. All of this does not provide very strong support for the claim that "GIA estimates depend on the used time period" (p. 21, ln 4). I would argue that as long as the input data is accurate, the time period shouldn't matter."*

We agree that as long as the input data is "correct" there is no time dependency. As you mentioned, the true limitations of the input data is the reason for differing results. The Multi-Mission-Altimetry product is dominated by CryoSat-2 observations during the time period 2010-07/2016-08. Fig. S1F shows the SEC. We decided not to use an additional CryoSat-2 only experiment. Fig. 2 (in this comment) compares Multi-Mission-derived and CryoSat-2-only-derived SEC. As you already mentioned, also Fig. 2 shows that the Envisat processing influences the result.
We rephrased the corresponding paragraphs in the Abstract, Sect. 5.5.

*"7) At several points in the paper, the authors present findings from a mixture of biased and dedebiased data sets. One example is in Sec 5.5 (p. 21, ln 4). Table S1 is another example. I can see the value in showing the magnitudes of the bias estimates, but a mass change result from, e.g., a debiased GIA solution and a biased GRACE solution, seems inconsistent. I would think you should only present either a fully biased or debiased solution to stay consistent. Otherwise, the various frame, deg1, and C20 biases get mixed differently depending on the combination chosen, and the solution becomes a mixture of global and regionally-constrained data."*

We fully agree and present in the revised version completely biased or debiased estimates only. We removed mixed values from the text (Sect. 2.4, 5.2, 5.5) and Table S1 from the supplementary material.

*"8) Modeling the elastic correction as a constant scale factor (pg 9, ln 15) of the altimetry height change may introduce error, especially in regions such as the AP and ASE (where thinning and accumulation are significant). Were the actual magnitudes of these elastic corrections investigated? And how would these BEC corrections be distinguished from large viscoelastic responses suspected in these same regions?"*

The constant scale factor does introduce error but this is negligible (Riva et al., 2009, Groh et al., 2012). The strong Gaussian smoothing further mitigates the influence of this error because large local amplitudes are damped. For illustration Fig. 3 (in this comment) compares vertical elastic deformation rates calculated from smoothed Multi-Mission-altimetry trends. This is done (1) by modelling in the spatial domain and (2) by the constant scale factor of 1.5 %. For (1) we used a predefined density mask to estimate mass change rates and rheological parameters from PREM. The differences between (1) and (2) vary between approximately -0.1 to 1.0 mm a$^{-1}$. In the AP only a small amount of altimetry observations can be used to determine elastic deformation. This introduces error, because true signal is presumably underestimated.

As we assume the viscoelastic deformation is purely linear we cannot separate it from the suspected truly non-linear signal in the ASE region.

We clarified the introduced error in Sect. 3.1 and extended the discussion in Sect. 5.1. The limitation through an assumed linearity of GIA is added in Sect. 2.5

*"9) While the authors are clear that it is a sensitivity study, there is no validation of the results (Gunter et al 2014 used GPS site displacements), so there is no assessment as to whether the variations observed by changing the input data sets are an improvement or not."*

As you mentioned, our aim is to address the sensitivity of an existing methodology from which we conclude limitations. The investigated approach might be inappropriate to judge the quality of input data sets with GNSS observations. All input data sets are combined with existing bias. This bias is jointly removed using the LPZ-based bias correction. During this step unknown systematic errors in the input data can cancel out each other. GNSS observations might judge those input data sets as an improvement.

**Minor items**

*"10) p. 12, ln 5: Just to clarify, the uncertainty you refere to here is the uncertainty as derived by the IMAU group for the mdot_firn term?"*

This is uncertainty we receive from our least square estimation when we estimate trends of cSMBA and taking 10 % of the estimated cSMBA trend over the ICESat-observation period.

We clarified this in Sect.4.1.

*"11) p. 12, ln 9: The 7-year window seems arbitrary. Why not 10 or 5 or 3 yrs? And, how does EOF analysis vary if another window timeframe is chosen?"*

We used a 7-year window, because this corresponds to the ICESat's observation period. We stated this in Sect. 3.3. The trend differences would increase if the time interval is shorter and decrease with a longer time window. Fig. 4 (in this comment) shows the first three EOFs using a 5 year, 7 year, and 10 year time interval, respectively. The dominant patterns remain with respect to spatial pattern and amplitude. Whereby the amount of the explained total variance changes.

*"12) p. 21, ln 4: certainly a longer period will result in more reliable results, but if the inputs are correct, the time period shouldn't matter. This statement should be rephrased to say that it depends on the quality of the input data sets."*

We agree and rephrased this statement in the Abstract and Sect. 5.5.

**Author comment to editor decision**

*"(1) The final sentence of reviewer 2's point 8 ("And how would these BEC corrections be distinguished from large viscoelastic responses suspected in these same regions?") is not fully addressed in your response. Please rectify this."*

We extended our comment (see above).

*"(2) Comparing figures 5 and 8, the difference in the sign of the GIA-induced BEC signal in the Siple Coast/Ross Sea region is striking. This is a region where the GIA-induced BEC signal is typically predicted to be positive. Please briefly comment on the reason for the sign change between the two figures."*

Fig. 5 (in this comment) shows smoothed cSMBA trends for the 2003-03/2009-10 period (corresponding to Fig. 5 in the manuscript) and for the 2010-07/2016-08 period (corresponding to Fig. 8 in the manuscript). During 2003-03/2009-10 and 2010-07/2016-08 time periods the sign of cSMBA and firn thickness trends over the Transantarctic Mountains is negative and positive, respectively. This is also partly visible in Fig. S1A and S1F in the supplementary material. We attribute the different sign in the GIA solutions to signal leakage from the Transantarctic Mountains into the Ross Sea region, due to filtering of high resolution input data.

[Figure]

Figure 1: Left: The differences between ICESat and FDM from input data sets we used. Right: The original figure from Gunter et al. (2014) with a threshold of 6 cm/a and the masked Kamb Ice Stream.

[Figure]

Figure 2: The comparison of Multi-Mission-Altimetry derived trends (left), CryoSat-2-only derived trends (middle), and the difference of Multi-Misson–CryoSat-2 (right). Note the different value range we choose to illustrate differences.

[Figure]

Figure 3: A: Trends from Multi-Mission altimetry (Gaussian smoothing with half response of 400 km). B: Therefrom derived elastic-induced bedrock elevation change using modelling in the spatial domain. The PREM earth model is used. C: Elastic deformation estimated with -1.5 % from A. D: The difference map between B and C.

[Figure]

Figure 4: Comparison of the EOF analysis using different time intervals. A–C, D–F, and G–I show the first three EOFs estimated over 5-year, 7-year, and 10-year time interval, respectively.

[Figure]

Figure 5: Smoothed cSMBA trends estimated over 2003-03/2009-10 (left) and 2010-07/2016-08 (right) time periods. A Gaussian filter with half-response of 400 km was applied.

[revised manuscript text omitted]

---

## Editor Decision (ED1)

I would like to thank the authors for addressing all comments on the previous version of the manuscript.

Many of the technical queries have now been resolved, but I list a number of minor points at the end of this document that require clarification. Page/line numbers refer to the most recently-uploaded 'track-change' version of the manuscript.

The grammar and punctuation of the manuscript are generally good, but the syntax of the text is occasionally awkward and I recommend detailed proof-reading by a native English speaker. In a few places your edits have resulted in ambiguities, or inconsistencies with the original text. One example is included in the list at the end of this document. Please check the logic of the text wherever you have made edits.

Two points require more detailed clarification:

1) Application of the elastic correction to altimetry observations: On page 4, line 8, you state that the elastic bed elevation change (BEC) term 'is reduced' – please explain what you mean by this (do you mean subtracted?) and clarify the reason for subtracting the elastic term from $\dot{\tilde{h}}_{alt}$ . Related to this, please define which of $\dot{\tilde{h}}_{alt}$ and $\dot{h}_{alt}$ is the observed quantity, and provide a physical interpretation for the other term. Finally, on page 9, line 14, you state that $\dot{\tilde{h}}_{alt}$ is scaled by 1.015 – does this scaling provide a revised value for $\dot{\tilde{h}}_{alt}$ or a value for $\dot{h}_{alt}$?

2) Application of the LPZ corrections: The steps taken to apply the 'LPZ-based GRACE bias correction' are unclear. From the manuscript (page 6, lines 4-5):

*"Prior to determining the mass-balance, a bias correction is applied to the total-mass change derived from time-variable gravity fields."*

However, from the 'author response' document:

*"GRACE-derived area density changes are not calibrated to the LPZ prior the actual combination (Eq. 9). GRACE-derived area density changes and the GIA solution from the combination are calibrated over the LPZ to determine the mass balance. In other words: The combined result derived from GRACE, altimetry and firn process models, namely the GIA-induced BEC, is calibrated over the LPZ".*

I suspect these statements are compatible, but a lot of the terminology used it not clearly defined, making it difficult to work out when or how the second bias correction is carried out, and which data sets are involved. In general, the whole of section 2.2 is difficult to follow - please review this section, and if necessary expand the text to clarify the details and motivation for the steps carried out.

Once these points and the issues below are addressed, I would be happy to review a revised version of the manuscript.

Kind regards,
Pippa Whitehouse

Minor technical points

- Page 1, line 17: "various time periods" – can you be more specific?

- Page 2, line 1: the edited sentence is unclear – you talk about mass balance being the difference between three things (an example of where an edit has led to confusion)

- Page 2, line 8: check the typical timescale of glacial cycle loading/unloading

- Page 2, line 35: You clarify on page 3 that you are using the term 'firn' to describe both SMB change and volume changes in the firn layer. However, before the reader reaches this statement, they are presented with the phrase "...**firn** processes, namely SMB and the volume change of the firn layer". I suggest editing this to "...**surface** processes, namely SMB and the volume change of the firn layer" to prevent any confusion at this point. Your definition of firn appears on the next line, after which it is fine to use your terminology.

- Page 5, line 9: you mention that the Kamb Ice Stream is treated separately, but you do not say how it is treated separately. Text on lines 14-15 hints at a mask being used for "regions of ice-dynamic thickening", but the regions are not specified. Please clarify.

- Page 5, line 11: "If the difference is not significant... it is not considered" – please clarify what is not being considered. Please also explicitly define what you mean by cases I, II and III, perhaps by using this terminology within equation 10.

- Page 21, lines 31-32: you refer to the assumption that GIA-induce BEC must be linear, but earlier (page 7, line 24) you acknowledge that this assumption may be violated under some conditions. Please consider whether the text on page 21 should be revised to reflect the information on page 7.

---

## Author Response (AR2)

We thank the editor Pippa Whitehouse for the review of the revised manuscript and for pointing to further limitations of the manuscript. We got help from a native English speaker who improved syntax and wording of the manuscript. Please find below our comments and the version of the revised manuscript which highlights differences to the previous version.

1) Application of the elastic correction to altimetry observations: On page 4, line 8, you state that the elastic bed elevation change (BEC) term 'is reduced' – please explain what you mean by this (do you mean subtracted?) and clarify the reason for subtracting the elastic term from  $\dot{\tilde{h}}_{alt}$ . Related to this, please define which of  $\dot{\tilde{h}}_{alt}$  and  $\dot{h}_{alt}$  is the observed quantity, and provide a physical interpretation for the other term. Finally, on page 9, line 14, you state that  $\dot{\tilde{h}}_{alt}$  is scaled by 1.015 – does this scaling provide a revised value for  $\dot{\tilde{h}}_{alt}$  or a value for  $\dot{h}_{alt}$ ?

 $\tilde{h}_{alt}$  is the quantity observed by altimetry containing all signals. This is summarised in Eq. 3. The elastic induced BEC through present-day ice mass change is comparatively small, but it needs to be subtracted from  $\dot{\tilde{h}}_{alt}$  before it is combined with the gravimetric observations. *A priori* the ice mass change is not known but it is necessary to estimate the elastic deformation. Practically, this load deformation can be roughly estimated by using altimetry observations itself. For this, we estimate  $\dot{h}_{elastic}$  with -1.5% of the altimetry observed SEC and subtract it. This is equal to scale the observations with 1.015 (Riva et al., 2009). The scaled quantity is  $\dot{h}_{alt}$  which is used for the combination (Eq. 8).

We made this clear in Sect. 2.1 and 3.1. We added Eq. 20 to summarise the elastic correction.

2) Application of the LPZ corrections: The steps taken to apply the 'LPZ-based GRACE bias correction' are unclear. From the manuscript (page 6, lines 4-5): "Prior to determining the mass-balance, a bias correction is applied to the total-mass change derived from time-variable gravity fields." However, from the 'author response' document: "GRACE-derived area density changes are not calibrated to the LPZ prior the actual combination (Eq. 9). GRACE-derived area density changes and the GIA solution from the combination are calibrated over the LPZ to determine the mass balance. In other words: The combined result derived from GRACE, altimetry and firn process models, namely the GIA-induced BEC, is calibrated over the LPZ". I suspect these statements are compatible, but a lot of the terminology used it not clearly defined, making it difficult to work out when or how the second bias correction is carried out, and which data sets are involved. In general, the whole of section 2.2 is difficult to follow - please review this section, and if necessary expand the text to clarify the details and motivation for the steps carried out. We agree that our description of the applied bias corrections is confusing. We apply the following processing steps:

- 1. Biased area density changes from GRACE ( $\dot{m}_{grav}$ ) and altimetry ( $\dot{h}_{alt}$ ) are combined to estimate the biased GIA signal ( $\dot{m}_{GIA}$ ).
- 2. The biased GIA signal is debiased using the *LPZ-based GIA bias correction* (Eq. 12).
- 3. The biased area density changes from GRACE are debiased using the *LPZ-based GRACE bias correction* (Eq. 14).
- 4. By combining 2. and 3. the debiased ice mass trend is estimated.

We extended Sect 2.2. by a step-by-step explanation.

**Minor technical points**

Page 1, line 17: "various time periods" – can you be more specific? We specified the time periods in the Abstract.

Page 2, line 1: the edited sentence is unclear – you talk about mass balance being the difference between three things (an example of where an edit has led to confusion) By revising the introduction we removed this paragraph.

Page 2, line 8: check the typical timescale of glacial cycle loading/unloading

We reformulated the corresponding sentence.

Page 2, line 35: You clarify on page 3 that you are using the term 'firn' to describe both SMB change and volume changes in the firn layer. However, before the reader reaches this statement, they are presented with the phrase "...**firn** processes, namely SMB and the volume change of the firn layer". I suggest editing this to "...**surface** processes, namely SMB and the volume change of the firn layer" to prevent any confusion at this point. Your definition of firn appears on the next line, after which it is fine to use your terminology.

We implemented this suggestion.

Page 5, line 9: you mention that the Kamb Ice Stream is treated separately, but you do not say how it is treated separately. Text on lines 14-15 hints at a mask being used for "regions of ice-dynamic thickening", but the regions are not specified. Please clarify. We added a technical explanation in the text.

Page 5, line 11: "If the difference is not significant... it is not considered" – please clarify what is not being considered. Please also explicitly define what you mean by cases I, II and III, perhaps by using this terminology within equation 10.

We edited the sentence and added cases I-III in Eq. 10.

Page 21, lines 31-32: you refer to the assumption that GIA-induce BEC must be linear, but earlier (page 7, line 24) you acknowledge that this assumption may be violated under some conditions. Please consider whether the text on page 21 should be revised to reflect the information on page 7. We added the possible non-linear deformation during 'short' periods in some regions at the end of Sect. 5.5.

**Sensitivity of inverse glacial isostatic adjustment estimates over Antarctica**

Matthias O. Willen1, Martin Horwath1, Ludwig Schröder1,3, Andreas Groh1, Stefan R. M. Ligtenberg2, Peter Kuipers Munneke2, and Michiel R. van den Broeke2

[revised manuscript text omitted]

$$\dot{m}_{\rm grav} = \dot{m}_{\rm GIA} + \dot{m}_{\rm ID} + \dot{m}_{\rm firn}.$$
(2)

15 Note that  $\dot{m}_{GIA}$  is not the GIA-induced mass trend: it is the apparent ADC because of the GIA-induced gravity-field changes. With ID all processes are summarised ID summarises all processes which are weighted with ice density, e.g. ice-dynamic flow or basal melt. We summarise the ice-induced, or cryospheric, area-density trend as  $\dot{m}_{ice} = \dot{m}_{ID} + \dot{m}_{firn}$ .

Analogously, the linear surface elevation change (SEC) overall linear SEC derived from altimetry  $h_{alt}$  is the sum of the linear SEC through ID, firn, GIA, and elastic BEC

20
$$\tilde{h}_{alt} = \dot{h}_{GIA} + \dot{h}_{elastic} + \dot{h}_{ID} + \dot{h}_{firm}.$$
 (3)

Note that GIA refers to the viscoelastic deformation of the solid Earth. The elastic BEC ( $\dot{h}_{elastic}$ ) through present-day ice-mass changes is reduced needs to be subtracted from the overall SEC observed by altimetry  $\dot{h}_{alt}$  prior to the combination by defining . We define  $\dot{h}_{alt} = \dot{h}_{alt} - \dot{h}_{elastic}$ . Doing this, the SEC signals in  $\dot{h}_{alt}$  are consistent with ADC signals in  $\dot{m}_{grax}$ .

The process-related elevation and area-density changes are linked with effective density assumptions ( $\rho_{GIA}$ ,  $\rho_{ID}$ )

| 25 | $\dot{m}_{ m GIA} =  ho_{ m GIA} \cdot \dot{h}_{ m GIA}$ | (4) |
|----|----------------------------------------------------------|-----|
|    | $\dot{m}_{ m ID} =  ho_{ m ID} \cdot \dot{h}_{ m ID}.$   | (5) |

Rearranging Eq. (3)

$$h_{\rm ID} = h_{\rm alt} - h_{\rm firn} - h_{\rm GIA}$$

(6)

and substituting it together with Eq. (4) and (5) into Eq. (2) leads to

$$\dot{m}_{\rm grav} = \rho_{\rm GIA}\dot{h}_{\rm GIA} + \rho_{\rm ID}(\dot{h}_{\rm alt} - \dot{h}_{\rm firn} - \dot{h}_{\rm GIA}) + \dot{m}_{\rm firn},\tag{7}$$

which can be solved for

$$\dot{h}_{\text{GIA}} = \frac{\dot{m}_{\text{grav}} - \rho_{\text{ID}}(\dot{h}_{\text{alt}} - \dot{h}_{\text{firm}}) - \dot{m}_{\text{firm}}}{\rho_{\text{GIA}} - \rho_{\text{ID}}}.$$
(8)

5

In Gunter et al. (2014), Eq. (8) is modified with a criterion to include assumptions about the difference  $\dot{h}_{alt} - \dot{h}_{firm}$  by *a priori* uncertainties.  $\rho_{ID}$  is replaced by  $\rho_{\alpha}$  to permit the following case distinction:

$$\dot{h}_{\rm GIA} = \frac{\dot{m}_{\rm grav} - \rho_{\alpha}(\dot{h}_{\rm alt} - \dot{h}_{\rm firn}) - \dot{m}_{\rm firn}}{\rho_{\rm GIA} - \rho_{\alpha}} \tag{9}$$

where

$$\rho_{\alpha} = \begin{cases}
\rho_{\text{ID}}, & \text{(I) if } \dot{h}_{\text{alt}} - \dot{h}_{\text{firm}} < 0 \\
& \text{and } |\dot{h}_{\text{alt}} - \dot{h}_{\text{firm}}| > 2\sigma_h \\
\rho_{\text{firn}}, & \text{(II) if } \dot{h}_{\text{alt}} - \dot{h}_{\text{firn}} > 0 \\
& \text{and } |\dot{h}_{\text{alt}} - \dot{h}_{\text{firn}}| > 2\sigma_h \\
0, & \text{(III) otherwise}
\end{cases}$$
(10)

10 with

$$\sigma_h = \sqrt{\sigma_{\dot{h}_{\text{alt}}}^2 + \sigma_{\dot{h}_{\text{fim}}}^2} \tag{11}$$

The case distinction is made to account accounts for uncertainties in altimetry and in the firn densification model (FDM) by using as well as *a priori* knowledge on ice-sheet processes. The GIA-induced BEC is in the millimetre per year range, whereas  $\dot{h}_{\text{firn}}$  and  $\dot{h}_{\text{ID}}$  can be in the centimetre to meter per year range. If altimetry and FDM are perfect,  $\dot{h}_{\text{alt}} - \dot{h}_{\text{firm}}$  would

- 15 leave essentially  $\dot{h}_{\rm ID}$  (apart from a very small  $\dot{h}_{\rm GIA}$ ). The following case distinction is made: If the altimetry-derived SEC is significantly more negative than SEC from the FDM, an ice-dynamic-induced SEC is assumed (glacial thinning). Gunter et al. (2014) argue that only one region in Antarctica is known to show glacial thickening: the area of the Kamb Ice Stream (Retzlaff and Bentley, 1993; Wingham et al., 2006). This region is therefore treated separately  $\therefore$  by a mask which sets  $\rho_{\alpha}$  to 917 kg m-3. The mask is generated from positive SEC from altimetry in this area. For case II in Eq. (10) it is assumed that the FDM
- 20 underestimates SEC due to firn processes and the remaining part therefore must not be weighted with ice density but with firn density. If the difference is not significant (smaller than  $2\sigma_h$ ), it is not considered this difference is ignored (case III in Eq. 10). In this case  $\dot{m}_{GIA} = \dot{m}_{grav} - \dot{m}_{firn}$  which means. That is, no mass change in the ice layer is considered. Mass changes and a mass trend of the ice sheet are fully described only arises by the trend of cumulated surface mass surface-mass-balance anomalies. This approach has the advantage to solve for GIA without a predefined spatial mask to distinguish between firn and

ice processes (e.g. density mask in Riva et al. (2009)) except for regions with ice-dynamic thickeningthe Kamb Ice Stream. An underestimated  $\sigma_h$  leads to differences between  $\dot{h}_{alt}$  and  $\dot{h}_{firn}$  being included in the mass balance, although they may not be significant. An overestimated  $\sigma_h$  will likely lead to case III in Eq. (10), also even for significant signals. In this case, data of altimetry and the model information of the FDM the altimetry data and FDM information are not taken into account – but and  $\dot{r}_h$  will be still fully update fully relied on

5  $\dot{m}_{\rm firn}$  and  $\dot{m}_{\rm grav}$  will be still fully used are fully relied on.

**2.2 Bias correction corrections and estimation of the mass balance**

**The estimation of (1) the GIA-induced BEC and (**

The following steps are performed in sequence:

- Step 1: Estimation of biased  $h_{GIA}$  using the data combination approach (Eq. 9)
- 10 Step 2: Removing the bias from  $\dot{h}_{GIA}$  leading to the debiased  $\tilde{h}_{GIA}$
  - Step 3: Removing the bias from  $\dot{m}_{grav}$  leading to the debiased  $\dot{\tilde{m}}_{grav}$
  - Step 4: Estimation of the debiased ice-mass trend from debiased apparent GIA-mass trend (Step 2) the mass balance is performed in a sequence. Gunter et al. (2014) crucially introduce two bias corrections and debiased total-mass trend (Step 3).

The bias corrections are necessary to consider offsets introduced e.g. by systematic errors in degree-1 and C20. The estimation

15 of the bias is done using the same strategy as Gunter et al. (2014). They argue that the effect of such offsets are significantly larger than potential mass signals in a low precipitation zone (LPZ) of the East Antarctic Ice Sheet. FirstIn Step 12, the LPZ-based GIA bias correction  $\dot{h}_{GIA,LPZ}$ -is applied. It is assumed that the GIA-induced BEC should be

negligibly small in this area. A remaining signal in the GIA estimate The GIA estimate from Step 1, averaged over the LPZ,  $\dot{\bar{h}}_{GIA,LPZ}$ , is interpreted as a bias due to the input data sets. Therefore the mean GIA-induced BEC within the LPZ  $\dot{\bar{h}}_{GIA,LPZ}$  is reduced It is subtracted from  $\dot{h}_{GIA}$ . The debiased GIA-induced BEC is

$$\dot{\tilde{h}}_{\text{GIA}} = \dot{h}_{\text{GIA}} - \dot{\bar{h}}_{\text{GIA,LPZ}}.$$
(12)

From which we derive the debiased apparent GIA-mass trend

20

$$\dot{\tilde{m}}_{\text{GIA}} = \tilde{\tilde{h}}_{\text{GIA}} \cdot \rho_{\text{GIA}}.$$
(13)

Input-data-set This means that input-data-set biases are jointly removed. The assumption of a negligible Removing a small
 GIA-induced BEC introduce error to introduces an error in the final result. GIA models predict approximately -3 to +1 mm a-1 in the area of the LPZ (Whitehouse et al., 2019). Gunter et al. (2014) argue that a the introduced error by the LPZ-bias correction is smaller than other bias contributionscontributors.

SecondIn Step 3, the LPZ-based GRACE bias correction  $\overline{m}_{grav,LPZ}$  is applied. Prior to determining the mass-balance, a bias correction is applied to the total-mass change derived from time-variable gravity fields. ADC from gravimetry are calibrated to

the LPZ by removing the mean ADC in this area,  $\dot{\bar{m}}_{\text{grav,LPZ}}$ . The debiased gravimetric ADC is

$$\dot{\tilde{m}}_{\text{grav}} = \dot{m}_{\text{grav}} - \dot{\bar{m}}_{\text{grav,LPZ}}.$$
(14)

The In Step 4, the debiased ice-mass trend is calculated as

$$\dot{\tilde{m}}_{\rm ice} = \dot{\tilde{m}}_{\rm grav} - \dot{\tilde{m}}_{\rm GIA}.$$
(15)

5 Note that the gravimetric bias correction is not applied to  $\dot{m}_{grav}$  used in Step 1, the initial combination (Eq. 9).

**2.3 Filtering**

A consistent spatial resolution of the data and models is required for the combination in the spatial domain. Moreover, a further noise suppression of GRACE-derived trends isrequired (Sect. For the necessary noise suppression we use GRACE data with a de-striping filter applied ( $\mathcal{F}_{DS}(\dot{m}_{grax})$ ) in addition to the filtering implied by the spherical harmonic truncation. Ideally, the data

- and models involved in the combination should have consistent spatial resolution, that is, they should be filtered consistently. 10 This is not strictly possible for the quotient  $(\dot{m}_{grav})/(\rho_{GIA} - \rho_{\alpha})$  in Eq. 3.2). Strictly speaking, only a filtered version of (9) because no unfiltered  $\dot{m}_{
[revised manuscript text omitted]

|                 |                                                                                                  |              | (incl. ERS-2, Envisat, ICESat, CryoSat-2)           |                                                 |  |  |  |

---

## Editor Decision (ED2)

**Editor's comments on revised version of TCD article "Sensitivity of inverse glacial isostatic adjustment estimates over Antarctica" by Willen et al.**

The authors' responses to the previous round of reviewer comments are clear, and all points previously raised have been resolved.

The authors have significantly improved the syntax and grammar of the manuscript following consultation with a native English speaker. However, a number of minor grammatical errors or ambiguities persist; these are listed at the end of this document.

Two sets of wider issues remain:

(1) I detail below a couple of queries regarding the definition of $\rho_\alpha$ which were missed in previous reviews

(2) The edits to the text have helped to clarify your arguments, but there are a few areas in the revised version of the Discussion that require further clarification

**Wider issues**

Regarding the definition of $\rho_\alpha$ in eq. 10: do the inequalities stated in cases (I) and (II) implicitly assume that $\dot{h}_{alt}$ is negative? If yes, then is $\rho_\alpha$ correctly defined in locations where $\dot{h}_{alt}$ is positive? Did you consider comparing the magnitude of the two terms? E.g. the first line of eq. 10 could be re-written as: "if $\left|\dot{h}_{alt}\right| - \left|\dot{h}_{firn}\right| < 0$" (although I note that if you compare the magnitudes then the sign of the inequalities in cases (I) and (II) may need to be reversed).

Definition of values assigned to $\rho_\alpha$: it seems odd to mention (on page 5, line 11, of manuscript-version4) that $\rho_\alpha$ is set to 917 kg/m³ across the Kamb Ice Stream because this is the same value that is used in all locations where $\rho_\alpha = \rho_{ID}$ (page 12, line 12). It would be useful if you could state the values used for all density constants in one place (most are not listed until section 3.4).

Terminology: You use the phrase 'apparent GIA-mass change' in several places, but occasionally the word 'apparent' is missing. Talking about 'GIA-mass change' is misleading given that the GIA-related process that is primarily detected by GRACE (and to a lesser degree, altimetry) is surface deformation - a process in which there is no net change in total mass within the footprint of the satellite observation. Consider using the phrase 'GIA-related mass change' in some parts of the text.

Section 5.1: on page 19 (lines 7-10) the wording of the text could be taken to imply that a negative value for GIA-induced BEC is unphysical. However, such a result is predicted across various regions of Antarctica via forward modelling, and is related to ice thickness increase since the Last Glacial Maximum due to well-documented accumulation increases. Please review the text with this in mind.

Bias correction assumptions: on page 20 you discuss the assumption that (lines 11-12) "over the LPZ the mean apparent GIA-mass change and the mean ice-mass change are zero". It is not clear from this text whether you assume that each of these terms is independently zero, or whether their sum (i.e. total mass change) is zero – please clarify. In this opening sentence to the paragraph, do you also need to mention that mean BEC is assumed to be zero across the LPZ?

**Minor grammatical issues and ambiguities**

Note: line numbers refer to manuscript-version4 and suggested edits are in *italics*. Suggestions are made with the intention of either correcting the grammar or helping to clarify the meaning of the text; they are not designed to enforce a particular writing style.

Page 2, line 15: the article by Caron and Ivins is listed as 'in press' in the reference list, please provide a doi if possible. The structure of the sentence containing this reference is awkward - please review.

Page 3, line 1: please define 'SMB' the first time this acronym is used

Page 3, lines 11-12: sentence is confusing, particularly the first half, please review

Page 4, line 9: suggest "The elastic BEC *triggered by* present-day ice-mass changes…"

Page 5, lines 8-9: suggest mentioning that the case described on these lines is case (I) in eq. 10

Page 5, lines 13-14: suggest combining two existing sentences: "If the difference is not significant (smaller than $2\sigma_h$) then $\rho_\alpha = 0$ (case III, eq. 10) and $\dot{m}_{GIA} = \dot{m}_{grav} - \dot{m}_{firn}$."

Page 5, line 16: please clarify whether "This approach" refers to case (III), which is discussed in the previous sentence, or the wider approach of using eq. 10 to define the value of $\rho_\alpha$

Page 5, line 19: please clarify what you mean by "significant signals"

Page 6, line 6: I think you mean "Step 2"

Page 6, line 14: suggest "the error introduced by…."

Page 8, line 1: "over decadal time scale" – check syntax

Page 8, line 2: it is not clear what you mean by "an adjusted trend"

Page 8, line 3: suggest "the viscoelastic deformation *associated with GIA* may be..."

Page 8, line 6: check the grammar in the opening sentence of section 2.6

Page 8, line 10: suggest "…is applied *to all data sets*"

Page 9, line 8: a comma is needed after "Further"

Page 10, line 3: suggest "…BEC of the solid Earth *due to* present-day…"

Page 10, line 7: suggest "*The* approximative nature…"

Page 10, line 29: suggest "…is adjusted to *fit* the filtered Stokes coefficients". A similar comment applies to lines 25-26 on page 11

Page 11, line 4: "containing" -> "which contains"

Page 11, line 19: check the use of commas in this line

Page 11, line 23: a comma is not needed after "implies"

Page 11, line 27: suggest "based on *the* ECMWF…"

Page 12, line 4: If you use the phrase, "the difference between…" you need to list two things

Page 12, line 6: "through" -> "associated with"

Page 12, line 11: "is by the" – check wording, are there some words missing?

Page 12, line 12: please define what you mean by "the GIA layer"

Page 12, line 24: please clarify what you mean by a "built-up difference"

Page 16, line 4: "to take account for a lower bound" – check syntax, meaning unclear

Page 16, line 8: typo - "density"

Page 16, line 20: suggest just "(d1_SLR), a range of…"

Page 18, line 2: "from" -> "of"

Page 18, line 6: suggest "values *of*…"

Page 18, line 17: "towards" -> "to"

Page 19, table 3 caption: suggest "…mass changes *calculated* in this study"

Page 19, line 4: "estimates" -> "estimate"

Page 19, line 12: suggest revising the sentence that begins "Even forward models…" to "*This issue cannot be resolved by considering the results of forward models because they also* show large variations *and sign differences* in the predicted spatial pattern of GIA-induced BEC."

Page 20, line 8: suggest "derived mass changes *are relative* to…"

Page 20, line 9: suggest re-ordering some text: "…and is made to ensure that the combination approach produces robust mass estimates."

Page 20, line 26: suggest "*These differences affect the results derived across* Kamb Ice Stream…"

Page 21, line 9: suggest "*an* ice-density weight"

Page 21, line 22: suggest "using" -> "that use"

Page 21, line 27: suggest "i.e. that *net* firn-thickness changes occur *over the modelling period*"

Page 21, line 30: suggest "The *variability in* these estimates…"

Page 22, line 3: suggest "In contrast" -> "However"

Page 23, lines 6-7: "14 years of used GRACE observations" – check syntax

Page 23, line 12: "eliminate" -> "eliminates"

Page 23, line 16: "over other time intervals" – please be a little more specific
* * *
**Pippa Whitehouse, 29th November 2019**

---

## Author Response (AR3)

We are grateful for the detailed review of the manuscript and the very helpful suggestions. The editor comments are indicated in italics. Blue and red text is used to indicate the author's response and changes in the manuscript, respectively.

**Wider issues**

*Regarding the definition of $\rho_\alpha$ in eq. 10: do the inequalities stated in cases (I) and (II) implicitly assume that $\dot{h}_{alt}$ is negative? If yes, then is $\rho_\alpha$ correctly defined in locations where $\dot{h}_{alt}$ is positive? Did you consider comparing the magnitude of the two terms? E.g. the first line of eq. 10 could be rewritten as: "if $|\dot{h}_{alt}| - |\dot{h}_{firn}| < 0$" (although I note that if you compare the magnitudes then the sign of the inequalities in cases (I) and (II) may need to be reversed).*

We do not implicitly assume that $\dot{h}_{alt}$ is negative. The case distinction accounts only for differences between $\dot{h}_{alt}$ and $\dot{h}_{firn}$ not for the individual magnitudes of $\dot{h}_{alt}$ and $\dot{h}_{firn}$. We agree that this was misleading in Sect. 2.1.

We revised the corresponding paragraph and clarified the case distinction in the text.

*Definition of values assigned to $\rho_\alpha$: it seems odd to mention (on page 5, line 11, of manuscript-version4) that $\rho_\alpha$ is set to 917 kg/m³ across the Kamb Ice Stream because this is the same value that is used in all locations where $\rho_\alpha = \rho_{ID}$ (page 12, line 12). It would be useful if you could state the values used for all density constants in one place (mos are not listed until section 3.4).*

We removed the particular density value from Sect. 2.1. Now we only indicate the density values in Sect. 3.4.

*Terminology: You use the phrase 'apparent GIA-mass change' in several places, but occasionally the word 'apparent' is missing. Talking about 'GIA-mass change' is misleading given that the GIA-related process that is primarily detected by GRACE (and to a lesser degree, altimetry) is surface deformation - a process in which there is no net change in total mass within the footprint of the satellite observation. Consider using the phrase 'GIA-related mass change' in some parts of the text.*

We used 'apparent' because the used GRACE processing projects all mass changes into one spherical layer (Eq. 1). GIA-induced mass changes occur in the subsurface which means they are not a process of area-density changes. Nevertheless, the apparent area-density change that would induce a gravity-field effect equal to the GIA-induced gravity-field effect can be put in relation to bedrock elevation change by the effective density $\rho_{GIA}$. For this reason, we called integrated GIA-related area density changes 'apparent GIA-mass change'. We agree that the phrase 'GIA-related mass change' makes this more clear.

We added a short explanation into Sect. 2.1 and used 'GIA-related mass change' instead of 'apparent GIA-mass change'

*Section 5.1: on page 19 (lines 7-10) the wording of the text could be taken to imply that a negative value for GIA-induced BEC is unphysical. However, such a result is predicted across various regions of Antarctica via forward modelling, and is related to ice thickness increase since the Last Glacial Maximum due to well-documented accumulation increases. Please review the text with this in mind.*

We agree and revised this paragraph.

*Bias correction assumptions: on page 20 you discuss the assumption that (lines 11-12) "over the LPZ the mean apparent GIA-mass change and the mean ice-mass change are zero". It is not clear from this text whether you assume that each of these terms is independently zero, or whether their sum (i.e. total mass change) is zero – please clarify. In this opening sentence to the paragraph, do you also need to mention that mean BEC is assumed to be zero across the LPZ?*

We clarified this.

**Minor issues**

We resolved all minor issues as suggested. We asked for the doi of the reference Caron and Ivins (2019) and added it to reference list. Note that this doi is not registered at the moment but will be as soon as the paper is published. Please find the revised manuscript below. Changes with respect to the previous version are highlighted.

[revised manuscript text omitted]

---

## Author Response (AR4)

Dear Pippa Whitehouse,

We are very pleased about this positive message and we revised all minor edits as suggested. Please find the revised manuscript below. Changes with respect to the previous version are highlighted.

Kind regards on behalf of all authors,
Matthias Willen

[revised manuscript text omitted]